# Hierarchical Diffusion for Efficient and Transferable Climate Downscaling

## Abstract

Downscaling is essential for generating the high-resolution climate data needed for local planning, but traditional methods remain computationally demanding. Recent years have seen impressive results from AI downscaling models, particularly diffusion models, which have attracted attention due to their ability to generate ensembles and overcome the smoothing problem common in other AI methods. However, these models typically remain computationally intensive. We introduce a Hierarchical Diffusion Downscaling (HDD) model, which introduces an easily-extensible hierarchical sampling process to the diffusion framework. A coarse-to-fine hierarchy is imposed via a simple downsampling scheme. HDD achieves competitive accuracy on the ERA5 reanalysis dataset and CMIP5 models, significantly reducing computational load by running on up to half as many pixels with competitive results. Additionally, a single model trained at 0.25° resolution transfers seamlessly across multiple CMIP5 models with much coarser resolution. HDD thus offers a lightweight alternative for probabilistic climate downscaling, facilitating affordable large-ensemble high-resolution climate projections; with a single model that can be applied across GCMs of varying input sizes. See a full code implementation at: https://github.com/HDD/HDD-Hierarchical-Diffusion-Downscaling.

## 1 Introduction

High resolution earth system data is critical for understanding and mitigating the impacts of anthropogenic climate change; however, generating it with traditional methodologies on a large scale is computationally prohibitive (Curran et al., 2024b) (Rampal et al., 2024). For example, general circulation models (GCMs) exhibit at least $\mathcal{O}(n^4)$ complexity with respect to resolution of the climate model due to the processing of variables in four dimensions (Balaji et al., 2022) (201, 2012) (Curran et al., 2024b). As a result, important historical and future climate datasets are often unavailable for use in local planning or only a limited subset of models is used, limiting the coverage of potential future climate outcomes(Riahi et al., 2017) (Rampal et al., 2024) (Curran et al., 2024b). In particular, for Australia, only two of the five Shared Socioeconomic Pathways (SSPs) from the IPCC's 6th Assessment Report have been downscaled to high resolutions, leaving large parts of the possible future scenario space unrepresented (Riahi et al., 2017).

In recent years, earth system modelling has undergone vast improvements owing to the proliferation of AI and advancements in computer science (Bi et al., 2022)(Lam et al., 2022)(Kochkov et al., 2024)(Curran et al., 2024a). Results from machine learning models in various earth-science tasks[1] are often competitive or superior on several metrics, but at a fraction of the inference cost (Hobeichi et al., 2023) (Bi et al., 2022)(Curran et al., 2024a) (de Burgh-Day and Leeuwenburg, 2023) . Downscaling, particularly, has seen promising advancements (Mardani et al., 2023), yet AI approaches are rarely compared against results from existing dynamical models and established climate metrics, which limits the trust that climate scientists can place in them.

Additionally, recent advances in computer vision have led to significant progress in auto-regressive image generation, with models such as VAR and GPT-4o demonstrating state-of-the-art performance (Tian et al.,

---

[1]Weather prediction, Downscaling, Climate Emulation and many more

2024) (Chen et al., 2025). These models exhibit favorable scaling properties and excel in generating high-fidelity images by modeling pixel or patch sequences auto-regressively. Despite their success, such approaches remain largely unexplored in weather and climate image generation, primarily due to the computational complexity and the challenges of capturing spatiotemporal consistency and physical realism required in geoscientific applications.

A small but growing subset of the literature has examined the concept of dimension destruction in diffusion models (Jin et al., 2024) (Zhang et al., 2022) (Campbell et al., 2023). In this approach, in addition to corrupting the training images with Gaussian noise, the dimension or size of the image is destroyed gradually. We present an easily extensible addition to the noise process which can be incorporated in existing diffusion models. By encouraging the model to learn at multiple resolutions, we construct a hierarchical schedule that downscales autoregressively from coarse to fine. We show that this framework can be easily applied to most existing diffusion model setups with some minor adjustments[2]. We also show that these models produce results competitive with, and in some cases surpassing, traditional dynamically downscaled models that nest Regional Climate Models (RCMs) within a GCM. These results are achieved over the Australian domain at a fraction of the inference and training cost.

1. We propose HDD (Hierarchical Diffusion Downscaling), a model that learns multi-scale representations via a hierarchical diffusion process. It applies dimension destruction with noise injection to enable robust feature learning across resolutions, followed by a coarse-to-fine reverse generation during inference. HDD is trained on varying spatial shapes to enforce scale consistency and improve high-resolution reconstruction.

2. Our proposed methodology is architecture-agnostic, allowing integration with any existing diffusion model by augmenting it with shape-conditioning resulting in resolution-aware conditioning mechanism. The framework supports plug-and-play usage within standard diffusion pipelines, making it broadly applicable to weather and climate models without architectural re-design.

3. We train HDD on ERA5 over the Australian domain and provide a comprehensive evaluation benchmark with climate metrics. The model passes all evaluation metrics and is competitive with other AI downscaling models and traditional dynamical RCMs, while requiring significantly less computational cost.

## 2  Related Work

Diffusion models implicitly generate images in a coarse-to-fine manner (Dielman, 2024) (Rissanen et al., 2022). This is consistent with the way humans process images, where coarse features are recognised first and finer details later; i.e. we 'see' the forest before the trees (Ho et al., 2020; Oliva and Torralba, 2006; Navon, 1977; Kauffmann et al., 2014).

Many atmospheric variables also exhibit these same power laws Willeit et al. (2014), analogous to the $1/f$ fractal spectra of natural images (van der Schaaf and van Hateren, 1996) (Hyvärinen et al., 2009). This indicates variability across scales, with dominant energy at larger scales but a continuous scale-invariant distribution down to finer scales. For example, the Nastrom-Gage spectrum, derived from global aircraft data, demonstrates a robust kinetic energy scaling from approximately 3000 km to several kilometers, following $k^{-3}$ at larger scales transitioning to $k^{-5/3}$ at smaller scales (Gage and Nastrom, 1986).

Regional analyses, including high-resolution reanalyses (e.g., ERA5 at 0.25°) and radar observations, also confirm this 1/f-like spectral behavior in atmospheric variables, such as precipitation intensity. These observations reveal continuous cascades from large storm systems down to small-scale showers without distinct breaks in scaling, consistent with findings from convection-permitting model forecasts in the U.S. (Gkioulekas and Tung, 2006).

The presence of self-similar, power-law spectra strongly motivates the application of coarse-to-fine multi-scale methods in atmospheric modeling and downscaling. Techniques like RainFARM exploit these scaling

---

[2]See methodology in section 3 for more details

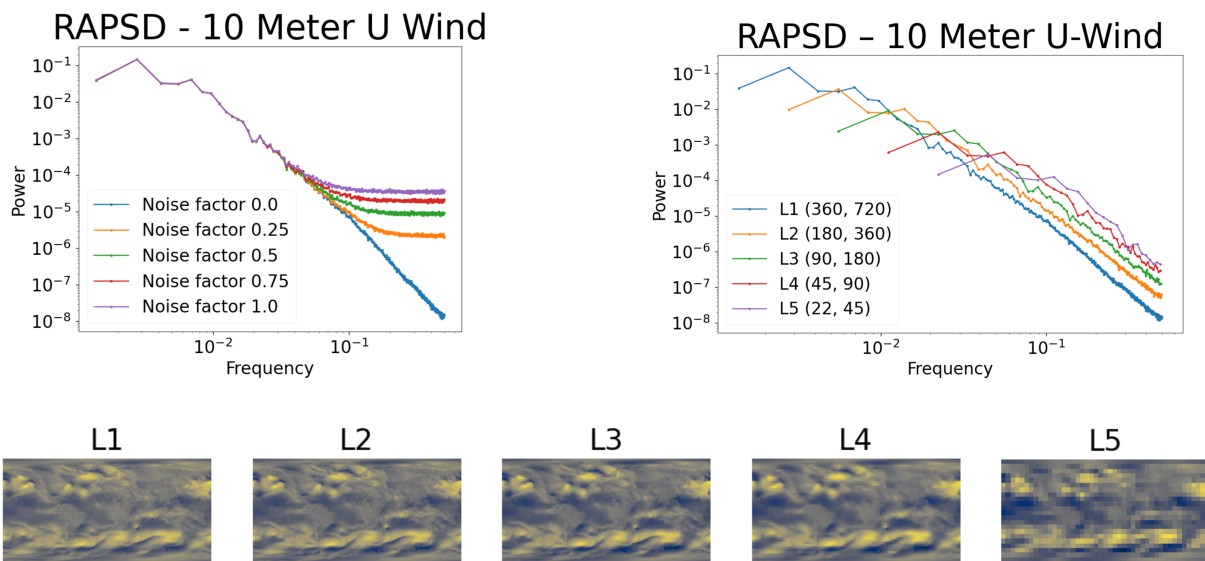

Figure 1: Radially Averaged Power Spectral Density (RAPSD) of the 10 metre U wind component. Left: High frequency finer features are the first to be corrupted by random noise. Conversely, the reverse diffusion process generates in a coarse-to-fine manner, contributing to the strong image generation capabilities of diffusion models. Enforcing this coarse-to-fine relationship explicitly can further improve results. Right: Weather data exhibits a clear power law. Coarser scales are much less information dense and closer to random noise. It is therefore, easier to model the coarser scales first, and progressively add higher frequency details. See appendix B for further details. Bottom: Progressively coarser wind data over grids of size 0.5 °(L1), 1.0 °(L2), 2.0 °(L3), 4.0 °(L4), 8 °(L5)

laws, generating fine-scale structures consistent with prescribed spectral properties, thereby reinforcing the conceptual and practical alignment between diffusion models in machine learning and traditional fractal-based atmospheric downscaling approaches (Rebora et al., 2006) (D'Onofrio et al., 2014).

## 3 Methodology

We impose an **explicit coarse-to-fine hierarchy** on the outputs of a noise-conditioned diffusion model by coupling the usual Gaussian noising process with a progressive *down-sampling / up-sampling* schedule that is sampled at every training step per the figure 4.

### 3.1 Baseline Elucidated Diffusion Model (EDM) formulation

We adopt a baseline implementation of EDM from (Watt and Mansfield, 2024) following Karras et al. (2022), which formalises and consolidates terminology from different diffusion methodologies. We refer the reader to the full paper (Karras et al., 2022) for further details. The basic setup is as follows:

Let $x_0 \in \mathbb{R}^{H \times W \times C}$ be a clean image (H: height, W: width, C: channels), and let $\{\sigma_t\}_{t=1}^{T}$ be a monotonically *increasing* noise schedule with $\sigma_0 = 0$ where $t = 1, \ldots, T$ represents the denoising timestep.

1. a *forward* (noising) kernel

$$q(x_t \mid x_{t-1}) = \mathcal{N}(x_t \,;\, x_{t-1}, \sigma_t^2 I), \qquad t = 1, \ldots, T,$$

2. and a learnable *reverse* kernel

$$p_\theta(x_{t-1} \mid x_t, \sigma_t) = \mathcal{N}(x_{t-1} \,;\, \mu_\theta(x_t, \sigma_t), \, \sigma_t^2 I),$$

where $\mu_\theta$ is the output of a score network trained to minimise the EDM loss.

### 3.2 Adding spatial hierarchies

Denote by $\mathbf{s}_t = (h_t, w_t)$ the *target resolution* at step $t$, with $(h_0, w_0) = (H, W)$ and $(h_T, w_T) \approx (1, 1)$. For each resolution we define

$$D_{\mathbf{s}_t} : R^{h_{t-1} \times w_{t-1} \times C} \longrightarrow R^{h_t \times w_t \times C}, \qquad U_{\mathbf{s}_t} : R^{h_t \times w_t \times C} \longrightarrow R^{h_{t-1} \times w_{t-1} \times C},$$

as bilinear *down-sampling* and matching *up-sampling* operators, respectively.

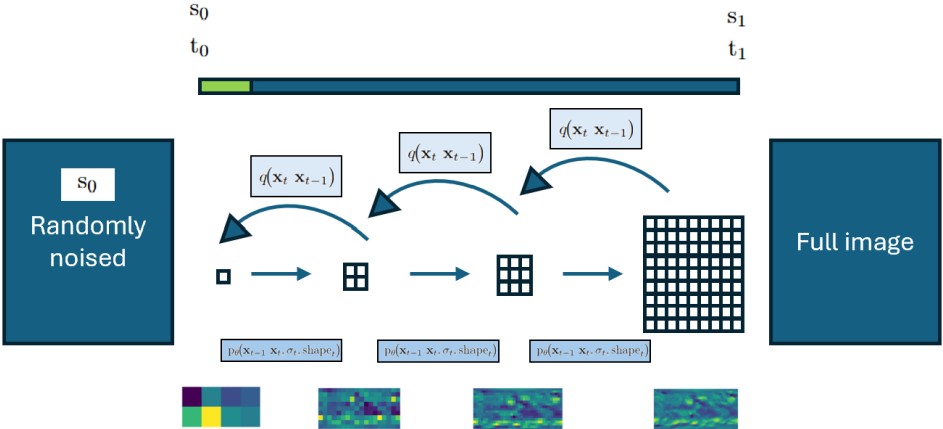

Figure 2: During training, the destruction kernel $q()$ progressively adds noise to the image and gradually destroys its dimensions according to the shape of the scheduler. At inference, the UNET $p()$ learns the inverse of this process, producing probabilistic downscaled outputs consistent with the original distribution. Note that although a UNET is used for $p()$ here, any function approximator could be applied. This hierarchical process is intuitive for image data, and in the climate setting we also show that competitive results can be achieved while processing up to half the pixels at inference.

**Hierarchical reverse process.** The network is trained to *simultaneously denoise and un-coarsen*. Conditioning on both $\sigma_t$ and the shape $\mathbf{s}_t$ we write

$$q(x_t \mid x_{t-1}) = \mathcal{N}(x_t ; x_{t-1}; \mathbf{s}_t, \sigma_t^2 I) \tag{1}$$

$$p_\theta(x_{t-1} \mid x_t, \sigma_t, \mathbf{s}_t) = \mathcal{N}(x_{t-1} ; \mu_\theta(x_t, \sigma_t, \mathbf{s}_t), \sigma_t^2 I), \tag{2}$$

Embedding every latent back to full resolution with $\tilde{x}_t = U_{\mathbf{s}_t}(x_t)$, the hierarchical EDM loss (HDD) is

$$\mathcal{L}_{\text{HDD}}(\theta) = \mathbb{E}_{t, x_0, \epsilon} \left[ w(\sigma_t) \left\| \epsilon - f_\theta(\tilde{x}_t + \sigma_t \epsilon, \sigma_t, \mathbf{s}_t) \right\|_2^2 \right],$$

where $f_\theta$ is the score network and $w(\sigma_t)$ is the usual EDM weighting term.

Because each $(h_t, w_t)$ is sampled once per example, the model learns a coarse-to-fine mapping with no additional passes through the data. Equations 1–2 reduce to the standard EDM when $D_{\mathbf{s}_t}$ and $U_{\mathbf{s}_t}$ are the identity[3], ensuring drop-in compatibility with existing code. The formulation aligns with the visual narrative in : blue arrows depict the noise and dimension-destruction kernels $q(x_t \mid x_{t-1})$, while straight arrows illustrate the learned reverse kernels $p_\theta(x_{t-1} \mid x_t, \sigma_t, \mathbf{s}_t)$.

---

[3]In this context, the identity would be if a shape schedule of the full final resolution is used throughout the whole schedule process: $s_t = (x_T, y_T)$ where $t = 1...T$ for each intermediate step

The algorithm in Table 1 outlines the sampling procedure for training and inference. Steps highlighted in blue are unique to the hierarchical model, which enforces a coarse-to-fine generation while exactly mirroring the usual practice of sampling the noise level $\sigma_t$ uniformly in log-space. Hence, the network encounters the full spectrum of noise levels *and* spatial resolutions while seeing each training sample only once.

---

**Algorithm 1**   Training (Hierarchical Forward Process)

---

**Require:** dataset $q(x_0)$, noise schedule $\{\sigma_t\}_{t=1}^T$, shape schedule $\{(h_t, w_t)\}_{t=1}^T$, network $f_\theta$

1: **repeat**
2:     $x_0 \sim q(x_0)$
3:     $t \sim \text{Uniform}(\{1, \ldots, T\})$
4:     $\epsilon \sim \mathcal{N}(0, I)$
5:     $x_t \leftarrow D_t(x_0)$    ▷ down-sample to $(h_t, w_t)$
6:     $z \leftarrow \sqrt{\bar{\alpha}_t}\, x_t + \sqrt{1 - \bar{\alpha}_t}\, \epsilon$
7:     **gradient step** on $\nabla_\theta \big\| \epsilon - f_\theta\big(U_t(z),\, t,\, (h_t, w_t)\big) \big\|_2^2$
8: **until** converged

---

**Algorithm 2**   Sampling (Hierarchical Reverse Process)

---

**Require:** $f_\theta$, noise $\{\sigma_t\}_{t=1}^T$, shape schedule $\{(h_t, w_t)\}_{t=1}^T$

1: $x_T \sim \mathcal{N}(0, I_{h_T \times w_T})$
2: **for** $t = T, \ldots, 1$ **do**
3:     $\tilde{x}_t \leftarrow U_t(x_t)$     ▷ upsample to full res
4:     $\epsilon_t \leftarrow f_\theta\big(\tilde{x}_t, t, (h_t, w_t)\big)$
5:     $x_{t-1} \leftarrow \frac{1}{\sqrt{\alpha_t}}\big(\tilde{x}_t - \frac{1-\alpha_t}{\sqrt{1-\bar{\alpha}_t}}\, \epsilon_t\big)$
6:     **if** $t > 1$ **then**
7:        $z \sim \mathcal{N}(0, I)$
8:        $x_{t-1} \leftarrow x_{t-1} + \sigma_t z$
9:     $x_{t-1} \leftarrow D_{t-1}(x_{t-1})$   ▷ project to next latent size
10: **return** $x_0$

---

Hence, the model first learns to minimise the coarse EDM loss and then progresively unlocks finer scales. We justify this two-phase optimisation strategy mathematically in Appendix G.

### 3.3   Coarse-to-Fine Diffusion Approximates the Score Function on Simpler Distributions

Recall that in standard diffusion models, the *score function* at time $t$ is

$$\nabla_x \log p_t(x),$$

which describes the gradient of the log-density for a progressively noisier version of $x_0$. Learning the reverse process is equivalent to learning to denoise (or equivalently approximate this score) at various noise levels.

**Coarse Distributions as Simpler Targets.**   When the image is downsampled to coarser resolutions (fewer pixels, fewer degrees of freedom), the induced data distribution

$$p_{\text{coarse}}(x) \;=\; \text{distribution of downsampled images}$$

is generally "simpler" to model. Intuitively, high-frequency details are removed, so spatial correlations are more tractable, and the manifold of coarse-resolution images is lower-dimensional. Consequently, predicting the score

$$\nabla_x \log p_{\text{coarse}}(x)$$

becomes easier: there are fewer fine-grained features to learn, and the model focuses on broad, low-frequency structure.

**Hierarchical vs. Single-Scale Approach.**   By first learning to approximate the score at a coarse distribution,

$$s_t^{(\text{coarse})}(x) \;\approx\; \nabla_x \log p_{\text{coarse},t}(x),$$

the model handles an easier inverse problem. Then, as the resolution is gradually increased, each subsequent diffusion (and corresponding score function) refines the result:

$$s_t^{(\text{finer})}(x) \;\approx\; \nabla_x \log p_{\text{finer},t}(x).$$

Each finer scale deals with distributions that are "closer" to the full-resolution distribution but still easier than jumping directly from pure noise to a full-resolution image in one step.

**Connection to Score-Based Diffusion.** In the continuous SDE view, we can think of downsampling as reducing the dimensionality or bandwidth of the data, so at time $t$, the *score* $\nabla_x \log p_t(x)$ lives on a simpler manifold. Discretizing this idea across multiple resolutions amounts to learning a sequence of denoising (or score) functions:

$$f_\theta\big(\mathcal{U}(x_t),\, t\big) \;\approx\; \nabla_x \log p_{\text{downsampled},t}(x),$$

where $\mathcal{U}(x_t)$ is an upsampled version of $x_t$. Such a hierarchical approach effectively breaks a complex score-estimation problem into stages where each stage handles a simpler, coarser distribution; with each increase in resolution progressively conditioning on the previous result. The autoregressive coarse to fine nature of this generation is intuitive–as previously discussed, humans understand images in a coarse to fine manner with

**Summary.** Hence, this method approximates the score function on these coarser distributions—where the image space is significantly reduced in size and complexity—before progressively shifting to higher resolutions. This *coarse-to-fine* strategy stabilizes training and provides a more direct way for the network to focus first on large-scale structure and then on finer details, thus approximating the score function in a stepwise manner from simpler (coarse) to more complex (full-resolution) distributions.

### 3.4 Why Increasing the Number of Dimension-Destruction Steps Improves Results

#### 3.4.1 Discretizing a Continuous Diffusion Process in Time

One perspective introduced in (Song et al., 2020), is to view diffusion models as discretized solutions to a *continuous-time* Stochastic Differential Equation (SDE). For standard DDPM (Ho et al., 2020) (without dimension destruction), the forward (noising) process in continuous time can be written as:

$$d\mathbf{x} = f(\mathbf{x}, t)\, dt \;+\; g(t)\, d\mathbf{w}, \tag{3}$$

where $\mathbf{w}$ is a standard Wiener process (Brownian motion) and $t \in [0, 1]$. Conceptually, $f(\mathbf{x}, t)$ and $g(t)$ define the drift and diffusion coefficients that gradually corrupt data into noise.

**Reverse-Time SDE.** By reversing the time variable from $t = 1$ down to $t = 0$, one obtains the *reverse* SDE:

$$d\mathbf{x} \;=\; \Big[f(\mathbf{x}, t) - g(t)^2\, \nabla_\mathbf{x} \log p_t(\mathbf{x})\Big] dt \;+\; g(t)\, d\overline{\mathbf{w}}, \tag{4}$$

where $p_t(\mathbf{x})$ is the (instantaneous) distribution of $\mathbf{x}$ at time $t$, and $\overline{\mathbf{w}}$ is a Brownian motion in reverse time. This reverse SDE perfectly "denoises" the corrupted data back to a clean sample as $t$ goes from 1 down to 0. We thank the reviewers for their identification of several limitations with this decomposition and discuss this further in Appendix K.

**Discrete Approximation via Euler–Maruyama.** In practice, we discretize the interval $[0, 1]$ into $N$ steps, $t_1 < t_2 < \cdots < t_N$. We define the discrete iterates $\{\mathbf{x}_{t_k}\}$ using an explicit numerical scheme (e.g., Euler–Maruyama):

$$\mathbf{x}_{t_{k-1}} = \mathbf{x}_{t_k} + \Big[f(\mathbf{x}_{t_k}, t_k) - g(t_k)^2\, \nabla_\mathbf{x} \log p_{t_k}(\mathbf{x}_{t_k})\Big] \Delta t + g(t_k)\, \sqrt{\Delta t}\, \boldsymbol{\eta}_k,$$

where $\Delta t = t_{k-1} - t_k$ and $\boldsymbol{\eta}_k \sim \mathcal{N}(0, \mathbf{I})$. While the update itself is an exact definition of the sequence, the resulting discrete path $\mathbf{x}_{t_k}$ serves as an approximation of the continuous-time process $\mathbf{x}(t_k)$.

**Time Discretisation** As $N \to \infty$, $\Delta t \to 0$, and the discrete chain converges to the exact solution of the SDE. Therefore, *more steps* $\implies$ *smaller local errors* $\implies$ *better final reconstruction*. Empirically, one observes that generating samples with more reverse steps (e.g., 1000 steps vs. 50) yields sharper images, because smaller increments in each denoising step incur fewer approximation artifacts (Ho et al., 2020).

### 3.4.2 Accumulation of Local Errors

Each discrete reverse step $p_\theta(\mathbf{x}_{t-1} \mid \mathbf{x}_t)$ can introduce some mismatch (e.g. KL divergence) compared to the true $q(\mathbf{x}_{t-1} \mid \mathbf{x}_t)$. If a single step has a small per-step error $\varepsilon$, over $N$ steps the total discrepancy might accumulate on the order of $O(N \cdot \varepsilon)$. However, when we *increase* $N$, the per-step error $\varepsilon$ often *decreases* because each denoising increment is smaller.

This can be made rigorous by considering the continuum limit $N \to \infty$, where $\Delta t = 1/N$. Classical SDE analysis (see (Kloeden and Platen, 1992)) shows that under certain regularity conditions, the global approximation error converges to zero as $\Delta t \to 0$. Thus,

$$\lim_{N \to \infty} p_\theta(\mathbf{x}_0) \ = \ q(\mathbf{x}_0),$$

meaning the learned model recovers the true data distribution in the idealized infinite-step regime (assuming perfect training).

Beyond discretisation, the total error in the generative process is bounded by three primary components Chen et al. (2022); Lee et al. (2022):

- **Initialization error**: which stems from the discrepancy between the terminal distribution of the forward SDE $p_T$ and the prior distribution $p_{prior}$ (usually $\mathcal{N}(0, \mathbf{I})$);

- **Score-matching (estimation) error**: reflecting the $L^2$-difference between the true score $\nabla \log p_t(\mathbf{x})$ and the learned model $s_\theta(\mathbf{x}, t)$; and

- **Discretisation error**: arising from the numerical solver (e.g., Euler-Maruyama) as discussed above.

Recent theoretical guarantees show that if the score error is small, the reverse SDE converges in KL divergence and Total Variation (TV) distance to the data distribution Song et al. (2020). Our method's coarse-to-fine approach effectively simplifies the score-matching task at early stages, potentially reducing the accumulation of errors across the sampling chain.

### 3.4.3 Dimension-Destruction Viewpoint (Coarse-to-Fine)

In the hierarchical shape-conditioning setting, we introduce a *second axis* of discretization: not only do we add noise at each step, but we also *downsample* (i.e. reduce the spatial resolution). Let us denote the forward dimension-destruction step at time index $k$ as

$$x_k \ = \ \mathcal{D}_k\Big(x_{k-1} + \epsilon_k\Big), \quad \epsilon_k \sim \mathcal{N}(0, \sigma_k^2 \mathbf{I}), \tag{5}$$

where $\mathcal{D}_k$ maps $(h_{k-1} \times w_{k-1})$ pixels to $(h_k \times w_k)$ pixels, typically $h_k < h_{k-1}$ and $w_k < w_{k-1}$. The reverse process then *upsamples* to full resolution before denoising. In effect, we are discretizing both (a) the *time* variable **and** (b) the *spatial dimension*.

**Finer Discretisation in Dimension.** When dimension changes are large (e.g. $64 \times 64 \to 4 \times 4$ in a single step), the model loses significant high-frequency content all at once, and the reverse step must hallucinate many details in one jump. This is akin to having a large $\Delta t$ in the SDE sense; the local error can be large and difficult to reverse.

Conversely, if we *split* that dimension change into multiple steps

$$(64 \times 64) \ \to \ (32 \times 32) \ \to \ (16 \times 16) \ \to \ (8 \times 8) \ \to \ (4 \times 4),$$

each step is a smaller "destruction" of high frequencies and structure, so the reverse upsampling + denoising is more accurate (less local error). By increasing the number of dimension-destruction steps, we make these changes more gradual, pushing the discrete approximation closer to a discrete limit in scale space:

$$\Delta(\text{dimension}) \ \to \ 0.$$

Hence, *more dimension-destruction steps* is loosely analogous to *more time steps* in standard DDPM: it yields finer increments, lower local error per step, and empirically a more faithful final reconstruction.

**Important caveat.** We emphasise that the parallel above between time-step refinement and dimension-step refinement is a *heuristic motivation*, not a formal convergence result. The Euler–Maruyama convergence argument cited in Section 3.4.1 holds for stochastic flows on a fixed-dimensional ambient space; our shape operator $\mathcal{D}_k$ changes the dimensionality of $x_k$ at every step, and the corresponding bilinear pair $(\mathcal{D}_k, \mathcal{U}_k)$ is non-invertible and discontinuous in $k$. We therefore make no claim of a continuous-time limit for the shape schedule, and the only formal guarantee we offer across scales is the KL chain-rule decomposition of Appendix D, which is a standard identity rather than a novel convergence theorem. The full statement of these limitations — including the fixed-dimension and non-adjoint-upsampling issues — is given in Appendix K. Modern convergence-bound literature relevant to the noise schedule (which *does* live in a fixed-dimensional ambient space after upsampling) is cited in (Chen et al., 2022; Lee et al., 2022) so that we do not re-derive standard results here.

Overall, we have two types of discretisation:

- **Noise-Level Discretisation:** Splitting the time interval $[0, 1]$ into small $\Delta t$'s (as in standard diffusion).

- **Dimension Discretisation:** Splitting the resolution reduction into many smaller downsampling increments.

Both can be justified under the same SDE-based argument: *smaller steps in each dimension of transformation $\implies$ lower approximation error per step $\implies$ better overall fidelity.* Empirical evidence (Tables in Section 5) corroborates that increasing either (or both) the number of time steps *and* dimension steps significantly improves generation quality metrics such as PSNR, SSIM, and FID.

## 4 What is the Theoretical Speedup?

We seek to define the theoretical improvement in processing speed for different shape schedules at training and inference. Note that as a standard UNET is being used as the underlying architecture, processing time scales linearly with the number of pixels; this would not be the case for an attention-based architecture like ViT, where processing time scales quadratically.

Note that this section represents the theoretical upper bound for performance improvement as this varies slightly with the size of image, the number timesteps T and ignores any overhead operations.

Let a clean image be $x_0 \in R^{H \times W \times C}$ with full area $A = H W$. During training we draw a *single* noise–shape index $t \sim \text{Uniform}\{1, \ldots, T\}$ per minibatch and replace the Gaussian–only corruption of EDM with the composite operator $x_{t-1} \mapsto D_{\mathbf{s}_t}(x_{t-1} + \sigma_t \varepsilon)$ that *downsamples first, denoises later.* Write $A_t = h_t w_t$ for the area at step $t$ and define the *normalised mean area*

$$\alpha \;=\; \frac{1}{T A} \sum_{t=1}^{T} A_t, \qquad \alpha \in (0, 1]. \tag{1}$$

$\alpha$ is the fraction of pixels—relative to baseline EDM—consumed *on average* by one network call; its reciprocal is therefore the ideal pixel/FLOP speed-up:

$$S_{\text{train}} \;=\; S_{\text{infer}} \;=\; \frac{1}{\alpha}. \tag{2}$$

We refer the reader to appendix H for the full proof but note that the hierarchical shape schedules have the following speedups:

| Shape scheduler | $\alpha$ | Speed-up $S = 1/\alpha$ |
|---|---|---|
| Linear shrink $(h_t, w_t) \propto 1 - \frac{t-1}{T-1}$ | $\frac{1}{3}$ | $3\times$ |
| Unit-shrink $(h_t, w_t) = (H - (t-1), W - (t-1))$ | see Eq. (3) | $\approx 1.32\times$ (50 steps) |

All schedules are *drop-in*: when $D_{\mathbf{s}_t} = U_{\mathbf{s}_t} = I$ they revert to vanilla EDM. Eq. equation 2 therefore gives an upper-bound on pixel, FLOP and memory savings obtainable with the HDD framework. We note that this is a higher speed up than comparable image-based approaches due to the choice of shape scheduler (Zhang et al., 2022).

# 5 Experiments

## 5.1 ERA5 Experiment

We trained a model on 30 years of ERA5 reanalysis data across temperature, u/v component of wind and precipitation from 1990 to 2019 over the Australian domain[4]. The task was to downscale the resolution from 1.5° to 0.25°. This was trained for 144 hours on two A100 GPUs for 360 epochs. Training time/resources were mimicked for Earth-ViT and the base EDM and all models achieved convergence. We then evaluate each of these models on the same task for five years of ERA5 data from January 2020 to December 2024 [5]. Note that Earth-ViT is based on the popular weather forecasting model panguweather with several slight modifications for the downscaling setting. See (Curran et al., 2024a) for further information on Earth-ViT and Appendix E for information on the training procedure for this setting.

Table 1: Performance of models trained on the ERA5 downscaling task over Australia, evaluated using Root Mean Squared Error (RMSE), Peak Signal-to-Noise Ratio (PSNR) and Continuous Ranked Probability Score (CRPS). Best performance for each metric shown in bold. The 'Base EDM' model corresponds to the backbone for popular downscaling model Corrdiff from NVIDIA (Mardani et al., 2023; Karras et al., 2022). The inclusion of different shape schedules is further examined in the ablation analysis in Section 5.2

| Model | RMSE | PSNR | CRPS |
|---|---|---|---|
| Bilinear Interpolation | 0.362755 | 9.54 | – |
| Earth-ViT (Finetuned PanguWeather (Curran et al., 2024a)) | 0.317410 | 10.63 | – |
| Base EDM - 500 steps | 0.000143 | 31.45 | 0.0002319 |
| HDD **(Ours)** | **0.000128** | **33.34** | **0.0002308** |

## 5.2 Ablation results - Hierarchical Scheduler

We now take the trained HDD model and examine the effect of different shape schedules to assess at what frequency dimensions should be increased versus just denoising. We take the best model on the ERA5 task (HDD - Hierarchical - 500 steps - 3 denoise steps per shape step) and reapply it with different shape schedules. As expected, there is a tradeoff between the total intermediate number of pixels processed at inference and RMSE. Interestingly, even the most extreme model, with equally spaced dimension jumps between (1,1) and (144,272), achieves very similar results to the base case. This suggests that passing additional shape information to the model enables it to capture coarse details similarly to a standard diffusion model, but at a fraction of the inference cost.

Interestingly, enforcing a slight coarse-to-fine generation with minimal overall pixel reduction appears to yield the best results. However, Different shape schedules show that increasing the number of denoising steps per shape (visualised in 3 as the flattening slope) leads to an initial improvement before leveling out. The final ablation '500 denoise steps per shape step' is equivalent to having no dimension-reducing steps, yet still performs much better than the base model in section 5.1. We hypothesise that by taking advantage

---

[4]Over a latitude/longitude bounding box of (-7.75,109) to (-43.5,176.75)

[5]See appendix E for further information on sampling procedures and ablations on metrics

Table 2: Same task as in section 5.1 but with differing shape schedules ablated. RMSE/PSNR are on the normalised residual.

| Model | RMSE | PSNR |
|---|---|---|
| Equally Spaced Steps | 0.000131 | 33.262 |
| Move in tandem – 1 denoise step per shape step | 0.000129 | **33.40** |
| 2 denoise steps per shape step | 0.000129 | 33.36 |
| 3 denoise steps per shape step **(Ours)** | **0.000128** | 33.34 |
| 50 denoise steps per shape step | 0.000131 | 33.21 |
| 500 denoise steps per shape step | 0.000133 | 33.11 |

of the hierachical nature of weather/climate data, the model is able to more efficiently model the coarser features in earlier diffusion steps. Additionally, the score function is able to more easily approximate the earlier coarsened steps, allowing the model to discretise the generation problem further. We expand on this reasoning and discuss several other hypotheses for this improved performance in Appendix A / B and Section 2 / 3.3.

## 5.3 Geographical Generalisation: European Domain

We also conducted an additional experiment over the European continent. The model was evaluated over a latitude/longitude bounding box of $(70.25°, -11.5°)$ to $(34.5°, 46.25°)$ using the identical experimental parameters outlined in Section 5.1.

As shown in Table 10, the HDD model yields significant improvements over the base Elucidated Diffusion Model (EDM) across all key metrics. It is critical to note that HDD shares the exact same underlying architecture as the base EDM; the performance gain is achieved purely through the addition of the two scalar inputs $(h_t, w_t)$ at inference. This highlights the robust spatial generalisation afforded by the hierarchical conditioning mechanism across diverse climatological regions.

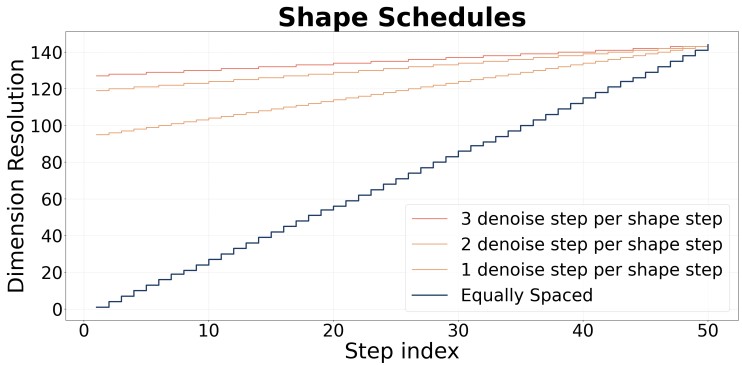

Figure 3: The dimensions of the downscaled image as it evolves with the number of steps. Note that in the extreme case where we equally our dimension jumps over the steps, we only process one third as many pixels at inference - see section four for further breakdown on this.

## 5.4 GCM Application and comparison to RCM simulations

We then applied the models trained in section 5.1 to precipitation simulations from multiple GCMs. GCM data is available at coarse resolution of approximately 1.5°, though finer and coarser resolutions also exist depending on the model.[6] grids due to the aforementioned computational constraints with generating this data[7]. We downscaled historical daily precipitation simulations from multiple GCMs: MIROC 5, CNRM-CM5, HadGEM and GFDL-ESM2M to 0.5° resolution over Australia and evaluated against precipitation observations from the Australian Gridded Climate Dataset (AGCD) (Jones, 1999).

The evaluation employs a set of minimum standard metrics focused on fundamental rainfall characteristics: total precipitation, spatial distribution, and seasonal cycle. We compute four metrics benchmarked against

---

[6]This equates to grids of approximately 167km x 167km
[7]See appendix B for further information

observations, with acceptance thresholds defined following (Isphording et al., 2024). The four metrics are detailed in appendix B alongside further details regarding this experiment..

Performance scores are displayed in table 4. Results show that both HDD and baseline EDM models achieve a pass across all four precipitation evaluation benchmarks. In comparison 20 out of 24 RCM simulations meet the benchmark criteria (Isphording et al., 2024).

Table 3: Relevant climate metrics SCorr, MAPE from (Isphording et al., 2024) and Computational Efficiency for MIROC5–driven downscaling runs

| Model | NRMSE | MAD | SCorr | MAPE | kgCO$_2$ Emitted |
|-------|-------|-----|-------|------|------------------|
| CCAM-1704 | 0.78 | 0.97 | 0.87 | 0.38 | ∼1032kg |
| Earth-ViT | 0.33 | 1.30 | 0.82 | 0.44 | ∼51kg Training + ∼2kg inference |
| Base EDM | 0.54 | 1.30 | 0.81 | 0.55 | ∼105kg Training + ∼5kg inference |
| HDD | 0.45 | 1.09 | 0.85 | 1.18 | ∼50kg Training + ∼2kg[8] inference |

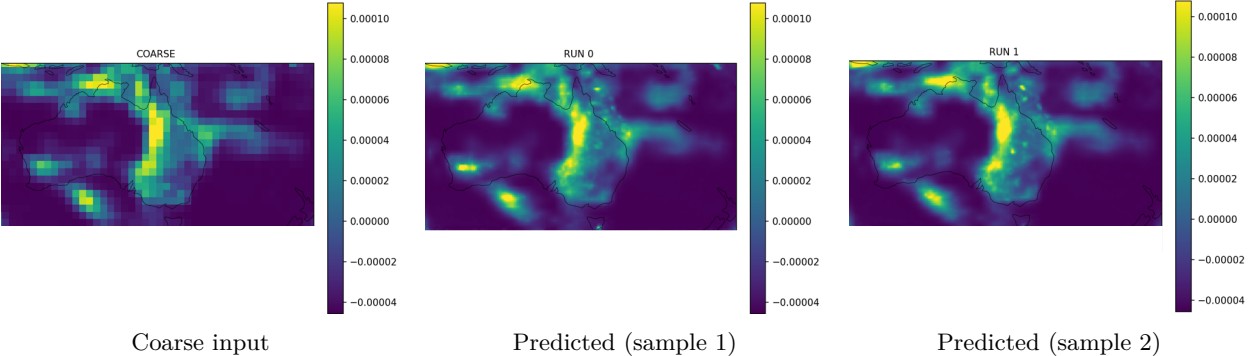

Coarse input · Predicted (sample 1) · Predicted (sample 2)

Figure 4: HDD applied over the Australian extent of the CNRM-CERFACS General Circulation Model. Leftmost panel shows the coarse GCM input; the two panels to the right are independent HDD ensemble draws conditioned on the same coarse input, illustrating the spread of plausible high-resolution outcomes produced by the probabilistic downscaler.

## 5.5 Resolution-agnostic capabilities of HDD

A practical requirement for downscaling is that a single model should transfer across General Circulation Models (GCMs) with *different native grid spacings*. We therefore evaluate HDD zero-shot across four CMIP5 GCMs spanning input resolutions from ∼1.4° to 2.5° × 2.0° and compare against the same backbone without hierarchies (EDM). Results below collate the precipitation metrics from (Isphording et al., 2024) (NRMSE, MAD in months, MAPE, spatial correlation).

**Limitation: MAPE degradation on heavy-tailed precipitation distributions.** We note that on MIROC5, CNRM-CM5 and HadGEM3, HDD's MAPE exceeds the Isphording et al. (2024) benchmark of $\leq 0.75$. This is also visible in Table 4 (MIROC5 MAPE 1.18; HadGEM3 MAPE 1.14). Our working hypothesis is that HDD's coarse-to-fine inductive bias preserves spatial structure well — which is consistent with the uniformly positive ΔSpCor across all four GCMs (+1.3% to +20.3%) — but that precipitation's heavy-tailed (approximately log-normal) distribution interacts poorly with bilinear downsample/upsample at every shape transition, because extreme values in the tail of the rainfall distribution are smoothed at each transition step. This would explain why HDD performs well on near-Gaussian fields (10u, 10v, 2t) but specifically degrades on `tp` for the GCMs whose precipitation distributions are heaviest-tailed (MIROC5, HadGEM3). The fact that GFDL-ESM2M — whose native resolution is coarsest and whose precipitation distribution is less heavy-tailed — shows +22.9% MAPE improvement is consistent with this hypothesis. Promising routes to recover MAPE include (i) log-transforming precipitation prior to the shape schedule, and

Table 4: Performance across MIROC5, CNRM-CM5, HadGEM, and GFDL-ESM2M. Best values for the downscaled benchmarks within each GCM group are highlighted in bold and second best are italicised.

| GCM | Resolution | Model | NRMSE | MAD | MAPE | SpCor |
|---|---|---|---|---|---|---|
| MIROC5 | 1.41°×1.41° | CCAM-2008 | 0.62 | **0.85** | 0.49 | **0.88** |
| | 1.41°×1.41° | CCAM-1704 | 0.78 | *0.97* | 0.38 | *0.87* |
| | 1.41°×1.41° | Earth-ViT | **0.33** | 1.30 | **0.44** | 0.82 |
| | 1.41°×1.41° | EDM | 0.54 | 1.30 | *0.55* | 0.81 |
| | 1.41°×1.41° | HDD | *0.45* | 1.09 | 1.18 | 0.85 |
| CNRM-CM5 | 1.406°×1.401° | CCAM-1704 | 0.68 | **0.87** | 0.54 | **0.82** |
| | 1.406°×1.401° | Earth-ViT | **0.50** | *1.07* | *0.30* | *0.79* |
| | 1.406°×1.401° | EDM | *0.56* | 1.27 | **0.25** | 0.78 |
| | 1.406°×1.401° | HDD | 0.62 | 1.39 | 0.57 | *0.79* |
| HadGEM | 1.875°×1.25° | CCAM-1704 | *0.78* | **0.95** | **0.27** | **0.89** |
| | 1.875°×1.25° | EDM | 0.94 | 1.90 | *0.54* | 0.76 |
| | 1.875°×1.25° | HDD | **0.49** | *1.26* | 1.14 | *0.85* |
| GFDL-ESM2M | 2.5°×2.0° | CCAM-2008 | **0.49** | **0.92** | *0.31* | **0.92** |
| | 2.5°×2.0° | CCAM-1704 | 0.69 | **0.92** | *0.31* | *0.90* |
| | 2.5°×2.0° | Earth-ViT | 0.61 | 1.15 | 2.51 | -0.13 |
| | 2.5°×2.0° | EDM | 0.73 | 1.35 | 0.35 | 0.50 |
| | 2.5°×2.0° | HDD | *0.51* | *0.96* | **0.27** | 0.71 |

Table 5: Paired comparison (HDD vs. EDM): percentage change by GCM/input resolution. **Sign convention (unified, per Reviewer 4FPr's request).** All entries are signed so that **positive = HDD improves on EDM** and **negative = HDD degrades relative to EDM**, regardless of whether the underlying metric is lower-is-better or higher-is-better. Concretely: for NRMSE/MAD/MAPE (lower better) we report $100\times(\text{EDM}-\text{HDD})/\text{EDM}$; for SpCor (higher better) we report $100\times(\text{HDD}-\text{EDM})/\text{EDM}$. Large-magnitude negative MAPE entries (e.g., MIROC5, HadGEM3) therefore indicate that HDD's MAPE *more than doubled* relative to EDM on those GCMs — this is explicitly flagged and discussed in the limitations paragraph below.

| GCM (input resolution) | ΔNRMSE (%) | ΔMAD (%) | ΔMAPE (%) | ΔSpCor (%) |
|---|---|---|---|---|
| MIROC5 (1.41°×1.41°) | +16.7 | +16.2 | −114.5 | +4.9 |
| CNRM-CM5 (1.406°×1.401°) | −10.7 | −9.5 | −128.0 | +1.3 |
| HadGEM3 (1.875°×1.25°) | +47.9 | +33.7 | −111.1 | +11.8 |
| GFDL-ESM2M (2.5°×2.0°) | +30.1 | +28.9 | +22.9 | +20.3 |

(ii) replacing bilinear with conservative remapping at shape transitions. HDD should therefore not be treated as a drop-in replacement for dynamical downscaling in precipitation policy applications until this is resolved, and we flag this in the broader-impact discussion.

**Analysis. (i) Spatial structure holds up as inputs get coarser.** HDD consistently *improves* spatial correlation over EDM across all GCMs (+1.3 to +20.3%), with the largest gain on the coarsest input (GFDL-ESM2M, +20.3% SpCor), indicating that hierarchical shape-conditioning helps reconstruct coherent fine-scale patterns even when the upstream grid is very coarse (Table 5).

**(ii) Error tradeoffs depend on the GCM and native resolution.** Relative to EDM, HDD often lowers distributional errors (MAPE: −114% to −128% on MIROC5/CNRM; −111% on HadGEM3), but can raise NRMSE/MAD for some models (e.g., HadGEM3 NRMSE +47.9%). This suggests HDD locks in where/when rainfall occurs (timing/structure) more reliably than exact amplitude for certain GCMs, which aligns with its coarse-to-fine inductive bias..

**(iii) Resolution-agnostic generalisation emerges from the hierarchy.** HDD was trained at a 1.5° input resolution but transfers to GFDL-ESM2M at 2.5°×2.0° with markedly higher SpCor than EDM (+20.3%) and competitive errors (Table 4). We hypothesise that the randomised shape schedules during training have taught the network to interpolate missing intermediate scales at inference, effectively acting as a learned multi-resolution prior; this makes it particularly suited to the downscaling problem where GCMs come in varying resolutions. We expand on this reasoning and discuss several other hypotheses for this improved performance in appendix A and Section 3.3.

**(iv) Positioning vs. non-probabilistic baselines.** Where available, Earth-ViT attains strong NRMSE on some GCMs (e.g., MIROC5), but (a) lacks native ensemble capability and (b) sometimes degrades spatial skill on very coarse inputs (GFDL, negative SpCor), whereas HDD keeps positive spatial skill and remains probabilistic for ensemble generation. This complements Section 5.1 where HDD matched or exceeded EDM at lower pixel budgets.

**Takeaways.** HDD preserves spatial coherence better than EDM as inputs get coarser; amplitude/timing errors show GCM-dependent tradeoffs that can be tuned by the shape schedule; the hierarchical conditioning confers practical resolution-agnostic behaviour, enabling a *single* trained model to service multiple GCMs of varying native grid size without re-architecture; and (iv) HDD retains the probabilistic benefits of diffusion (ensembles, uncertainty) while operating on fewer pixels (cf. Section 4).

## 6 Conclusion

Overall, HDD demonstrates that a coarse-to-fine ladder inside a diffusion model reduces the pixel budget - and therefore the FLOPs and $CO_2$ - by roughly two-thirds, while preserving or even improving performance when tested on ERA5 and historical simulations from CMIP5 GCMs. Regional climate fields are produced at a fraction of the computational and monetary cost of both dynamical downscaling (RCM simulations) and standard diffusion models, with performance comparable to dynamical downscaling. The method is architecture-agnostic and applicable to any diffusion framework, although retraining is needed to incorporate two additional scalar inputs[9]. HDD can also be extended to a multi-model ensemble, enabling the generation of large ensembles of regional climate projections and providing a pathway to assess the full spectrum of possible future climate outcomes, which is currently underrepresented. Although we note that the model is not tested on future GCM data, this could be an interesting next direction for this work. [10] Finally, HDD and other AI-based downscaling methods can viewed as an alternative that complements dynamically downscaled results, expanding the range of feasible ensemble sizes and scenario coverage while reducing computational cost.

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

# A    Radially-Averaged Power-Spectral Density (RAPSD)

Figures 1 and 2 in the main text visualise how power is redistributed when an ERA5 field is either (i) coarsened by repeated $2\times$ down-sampling or (ii) contaminated with Gaussian noise of increasing variance. This appendix formalises the metrics that underlie those plots and derives the theoretical curves that explain their shapes.

## A.1    From a 2-D field to a 1-D spectrum

Let the clean field be a real-valued array $x \in \mathbb{R}^{N_y \times N_x}$, indexed by pixel coordinates $\mathbf{r} = (n, m)$. Its discrete Fourier transform (DFT) is

$$X(\mathbf{k}) \;=\; \mathcal{F}\{x\}(\mathbf{k}) \;=\; \sum_{n=0}^{N_y-1} \sum_{m=0}^{N_x-1} x_{n,m}\, e^{-2\pi i \left(k_y n/N_y + k_x m/N_x\right)}, \quad \mathbf{k} = (k_x, k_y). \tag{6}$$

The *power-spectral density* (PSD) is the squared magnitude $P(\mathbf{k}) = |X(\mathbf{k})|^2$. For many geophysical or photographic images the statistics are approximately isotropic, so it is convenient to collapse the 2-D PSD into a 1-D function of radius $f = \|\mathbf{k}\|$ (spatial frequency):

$$\mathrm{RAPSD}(f_j) \;=\; \frac{1}{|\mathcal{A}_j|} \sum_{\mathbf{k} \in \mathcal{A}_j} P(\mathbf{k}), \quad \mathcal{A}_j = \left\{ \mathbf{k} : f_j \leq \|\mathbf{k}\| < f_{j+1} \right\}, \tag{7}$$

where the annuli $\{\mathcal{A}_j\}_{j=0}^{J-1}$ partition the Fourier plane into logarithmically-spaced bins ($f_{j+1}/f_j = \mathrm{const}$). Log–log plots of $f \mapsto \mathrm{RAPSD}(f)$ often reveal the empirical power law $\mathrm{RAPSD}(f) \propto f^{-\alpha}$ with $\alpha \approx 2$ for natural images and for many meteorological fields (e.g. the Nastrom–Gage kinetic-energy spectrum).

## A.2    Effect of 2 x spatial down-sampling

Down-sampling by an integer factor $s$ reduces the Nyquist frequency from $f_{\max} = 1/2$ (in pixel units) to $f_{\max}/s$. Assuming ideal low-pass pre-filtering, all energy above $f_{\max}/s$ is discarded; below that limit the PSD is merely scaled by $s^2$ to conserve total variance. For five binary-decade reductions $s \in \{2, 4, 8, 16, 32\}$ (as in **Fig. 1**) one expects

$$\mathrm{RAPSD}_s(f) \;=\; \begin{cases} s^2\, \mathrm{RAPSD}_{\mathrm{orig}}(f), & f < \frac{1}{2s}, \\ 0, & f \geq \frac{1}{2s}. \end{cases}$$

Hence the curves in Fig. 1 coincide at low $f$ and peel off successively at their (progressively smaller) Nyquist cut-offs—exactly the trend observed.

## A.3    Effect of additive white noise

Let $\varepsilon \sim \mathcal{N}(0, \sigma_n^2)$ be i.i.d. pixel noise. Because the DFT is linear and because white noise is spectrally flat,

$$Y = X + \varepsilon, \tag{8}$$

$$\mathbb{E}\left[P_Y(\mathbf{k})\right] = P_X(\mathbf{k}) + \sigma_n^2, \tag{9}$$

so every RAPSD curve is translated upward by the *same* constant $\sigma_n^2$. Define the *hinge frequency* $f^\star(\sigma_n) = \min\{f : \mathrm{RAPSD}_X(f) \leq \sigma_n^2\}$. For $f < f^\star$ the spectrum is signal-dominated and remains unchanged; for $f > f^\star$ the spectrum is noise-dominated and becomes flat at the level $\sigma_n^2$. Because the notebook sets $\sigma_n = \mathrm{noise\_factor} \times \sigma_X$, the plateau height scales quadratically with the chosen `noise_factor`. The five coloured curves of **Fig. 2** therefore realise

$$\mathrm{RAPSD}_{\mathrm{noise}}(f \mid \lambda) \;=\; \mathrm{RAPSD}_X(f) + \lambda^2\, \sigma_X^2, \quad \lambda \in \{0, 0.25, 0.5, 0.75, 1.0\}.$$

Each ten-fold rise in $\lambda^2$ shifts the horizontal tail up by the same factor and pushes $f^\star$ leftward, reducing the bandwidth in which the underlying flow field is recoverable.

### A.4 Relevance for coarse-to-fine diffusion scheduling

In a Gaussian diffusion process the state at noise level $\sigma_t$ is $x_t = \alpha_t x_0 + \sigma_t \varepsilon$ (with $\varepsilon \sim \mathcal{N}(0,1)$). Because RAPSD adds linearly, low frequencies re-emerge first as $\sigma_t$ decays, while high-frequency detail appears only when $\sigma_t^2 \lesssim \mathrm{RAPSD}_X(f)$. Thus the reverse-time sampler automatically follows a coarse-to-fine trajectory in spectral space. The patterns in Figures 1 and 2 therefore provide the mathematical justification for conditioning a diffusion model either on a sequence of coarsened resolutions or on a carefully curated noise schedule when generating meteorological fields.

## B Climate Benchmark Metrics

This appendix concisely defines the minimum-standard metrics proposed by (Isphording et al., 2024) and adopted in the present study to assess the ability of ML models to capture three fundamental characteristics of rainfall: *How much does it rain? Where does it rain?*, and *When does it rain?* The metrics used to quantify these characteristics are listed in Table 2 below. Each metric is accompanied by ($i$) its mathematical formulation as implemented in our analysis code, ($ii$) the numerical benchmark that constitutes a 'pass', and ($iii$) a brief explanation of what the metric tells us from a climate science perspective. Where relevant, $n$ denotes the number of non-missing land grid-cells after the AGCD quality mask is applied and $w_i$ the cosine-latitudinal area-weight for cell $i$. $P_i$ and $O_i$ are the predicted and observed climatological-mean annual rainfall total respectively. We also apply these metrics to evaluate three dynamically downscaled precipitation simulations produced CNRM-CM5 CCAM-1704, HadGEM2-ES RegCM4-7, and MIROC5 CCAM-1704. These datasets are part of the CORDEX-CMIP5 ensemble over the Australasian domain.

Table 6: Downscaling performance metrics and estimated carbon footprint for machine-learning (ML) and dynamical-downscaling (DD) configurations over Australia (1976–2005). Results for both HDD and the base EDM were both performed using 50 inference steps and 3 denoise steps per shape step for the HDD models

| Driving GCM | Model / RCM | NRMSE | MAD | SCorr | MAPE | kgCO$_2$ Train [11] |
|---|---|---|---|---|---|---|
| MIROC5 | HDD | 0.45 | 1.0865 | 0.85 | 1.1752 | ~53 kg |
| CNRM-CM5 | HDD | 0.62 | 1.3897 | 0.79 | 0.5185 | ~53 kg |
| MIROC5 | Base EDM | 0.54 | 1.2961 | 0.81 | 0.5505 | ~105 kg |
| CNRM-CM5 | Base EDM | 0.56 | 1.2654 | 0.78 | 0.2481 | ~105 kg |
| CNRM-CM5 | Earth-ViT | 0.50 | 1.1044 | 0.80 | 0.3498 | ~51 kg |
| CNRM-CM5 | CCAM-1704 | 0.68 | 0.8698 | 0.89 | 0.2600 | 1032 kg (total) |
| HadGEM2-ES | RegCM4-7 | 1.47 | 1.1586 | 0.90 | 1.0223 | 588 kg (total) |
| MIROC5 | CCAM-1704 | 0.78 | 1.1083 | 0.88 | 0.3374 | Not Available |

### B.1 Mean Absolute Percentage Error (MAPE) - How much does it rain?

$$\mathrm{MAPE} = \frac{1}{W} \sum_{i=1}^{n} w_i \left| \frac{P_i - O_i}{O_i} \right|, \qquad W := \sum_{i=1}^{n} w_i. \tag{10}$$

**Benchmark:** MAPE $\leq 0.75$.

*Climate meaning.* MAPE gauges the proportional bias in climatological mean annual rainfall: values $\leq 0.75$ require simulations to be, on average, within 75% of observations—a pragmatic trade-off between model realism and current skill levels.

### B.2 Spatial Correlation (SCor) - (Where does it rain?)

$$\mathrm{SCor} = \frac{\sum_{i=1}^{n} w_i, (P_i - \hat{P})(O_i - \hat{O})}{\sqrt{\sum_{i=1}^{n} w_i, (P_i - \hat{P})^2} \sqrt{\sum_{i=1}^{n} w_i, (O_i - \hat{O})^2}}, \qquad \hat{P} := \frac{1}{W} \sum_{i=1}^{n} w_i P_i, \quad \hat{O} := \frac{1}{W} \sum_{i=1}^{n} w_i O_i. \tag{11}$$

Table 7: Minimum-standard rainfall metrics (adapted from (Isphording et al., 2024)). Metrics are computed from area-weighted average total rainfall over Australia using the AGCD observational dataset. Amplitude is the difference between maximum and mean monthly rainfall; phase is the month of maximum rainfall.

| Fundamental rainfall characteristic | Quantifying metric | Benchmark threshold |
|---|---|---|
| How much does it rain? | Mean absolute percentage error (MAPE) | MAPE $\leq 0.75$ |
| Where does it rain? | Spatial correlation (SCor) | SCor $\geq 0.7$ |
| When does it rain? | *Amplitude*: normalised root mean squared error (NRMSE) 
 *Phase*: mean absolute deviation (MAD; months) of the maximum-rainfall month | *Amplitude*: NRMSE $\leq 0.6$ 
 *Phase*: MAD $\leq 2$ |

where $\hat{P}$ and $\hat{O}$ are the area-weighted spatial means of $P_i$ and $O_i$.

**Benchmark:** SCor $\geq 0.7$.

*Climate meaning.* SCor evaluates how well the model reproduces the *spatial pattern* of mean annual rainfall. correlations ($\geq 0.7$) imply that regional wet and dry zones are captured in the right places, even if the absolute totals differ.

### B.3   Seasonal-cycle metrics (When does it rain?)

We define the grid-cell seasonal amplitude $A_i$, where $A_i = P_i^{\mathrm{max}} - \bar{P}_i$, and $M_i$ as the phase (month index) of that maximum $P_i^{\mathrm{max}}$. The Normalised RMSE of amplitudes:

$$\mathrm{NRMSE} = \frac{\sqrt{\frac{1}{W}\sum_i w_i (A_i^{\mathrm{mod}} - A_i^{\mathrm{obs}})^2}}{\frac{1}{W}\sum_i w_i, A_i^{\mathrm{obs}^2}} \tag{12}$$

where $A_i^{\mathrm{mod}}$ and $A_i^{\mathrm{obs}}$ are the amplitudes of the prediction and observation respectively.

**Benchmark:** NRMSE $\leq 0.60$.

$$\mathrm{MAD} = \frac{1}{W}\sum_{i=1}^{n} w_i \big| \Delta\phi(M_i^{\mathrm{mod}}, M_i^{\mathrm{obs}}) \big| \tag{13}$$

where $\Delta\phi$ is the shortest circular distance on a 12-month circle. $M_i^{\mathrm{mod}}$ and $M_i^{\mathrm{obs}}$ are the predicted and observed phases respectively.

**Benchmark:** MAD $\leq 2$.

## C   Calculations for Carbon Emitted into Atmosphere from Model Training/Inference

### C.1   Key inputs and assumptions

- **Service-unit (SU) charging on HPC software.** CPU queues are charged at $2\,\mathrm{SU\,core^{-1}\,h^{-1}}$, the V100 queue at 3 SU per "resource*hour", and the A100 queue at $4.5\,\mathrm{SU\,resource^{-1}\,h^{-1}}$.

- **Hardware power draw.**
    - *CPU node* $P_{\mathrm{system}} = 2.90\,\mathrm{MW}$ , i.e. $P_{\mathrm{core}} = 14.2\,\mathrm{W}$.

- *DGX A100 node.* $P_{\mathrm{node}} = 6.5$ kW for 8×A100 GPUs[12] $\Rightarrow P_{\mathrm{GPU}} = 0.81$ kW.
- *DGX-1 V100 node.* Thermal-design power $P_{\mathrm{node}} = 3.2$ kW for 8×V100 GPUs $\Rightarrow P_{\mathrm{GPU}} = 0.40$ kW.

- **Data-centre overhead.** Power-usage-effectiveness (PUE) assumed PUE $= 1.3$[13].

- **Grid-emission factor** Scope-2 factor $0.68\,\mathrm{kg\,CO_2\,kWh^{-1}}$ and scope-3 factor $0.05\,\mathrm{kg\,CO_2\,kWh^{-1}}$ from the

  Combined factor used below: $\gamma = 0.73\,\mathrm{kg\,CO_2\,kWh^{-1}}$ .

## C.2 CPU workloads

**Energy per core-hour** $E_{\mathrm{core}} = P_{\mathrm{core}} \times \mathrm{PUE} = 14.2\,\mathrm{W} \times 1.3 = 18.5\,\mathrm{W} = 0.0185$ kWh.

**SU–to–energy conversion** CPU: 2 SU per core-hour, hence $E_{\mathrm{SU}} = 0.0185\,\mathrm{kWh}/2 = 9.25 \times 10^{-3}$ kWh.

**Carbon per kSU** $1\,\mathrm{kSU} = 1\,000\,\mathrm{SU} \Rightarrow m_{\mathrm{kSU}} = 1\,000\,E_{\mathrm{SU}}\,\gamma = 1\,000 \times 9.25 \times 10^{-3} \times 0.73 \approx 6.8\,\mathrm{kg\,CO_2\,e}$.

## C.3 GPU workloads

### A100

$$E_{\mathrm{GPU\ h}} = P_{\mathrm{GPU}} \times \mathrm{PUE} = 0.8125\,\mathrm{kW} \times 1.3 = 1.06\,\mathrm{kWh},$$
$$m_{\mathrm{GPU\ h}} = E_{\mathrm{GPU\ h}}\,\gamma \approx 1.06 \times 0.73 = 0.77\,\mathrm{kg\,CO_2\,e}.$$

*Six-hour run on a single A100:* $m = 6 \times 0.77 \approx 4.6\,\mathrm{kg\,CO_2\,e}$.

### V100

$$E_{\mathrm{GPU\ h}} = 0.40\,\mathrm{kW} \times 1.3 = 0.52\,\mathrm{kWh},$$
$$m_{\mathrm{GPU\ h}} = 0.52 \times 0.73 = 0.38\,\mathrm{kg\,CO_2\,e}.$$

*One hour per V100:* $0.38\,\mathrm{kg\,CO_2\,e}$.

## C.4 Summary

| Workload | Energy $(\mathrm{kWh\,h^{-1}})$ | $CO_2$ $(\mathrm{kg\,h^{-1}})$ |
|---|---|---|
| CPU | 0.0185 | 0.014 |
| A100 GPU | 1.06 | 0.77 |
| V100 GPU | 0.52 | 0.38 |

These values were scaled for the various runtimes involved in each script.

# D Chain-rule proof of the KL–decomposition

Note that the terminology and logic here is adapted from (Rudin, 1987) (Royden and Fitzpatrick, 2010) but has been adapted for the downscaling setting.

Throughout we write $\lambda_d$ for the *d-dimensional Lebesgue measure*, (i.e. the ordinary notion of volume for a distribution in $\mathbb{R}^d$ - we only define this here to get some nice continuity guarantees for our marginals later)

---

[12]See A100 power draw per nvidia specifications: https://www.nvidia.com/content/dam/en-zz/Solutions/Data-Center/nvidia-dgx-a100-datasheet.pdf
[13]Specific to local cluster

Formally, a probability law $q$ on $\mathbb{R}^d$ is said to be *absolutely continuous* with respect to $\lambda_d$, written $q \ll \lambda_d$, if there exists a non-negative integrable function $q(x)$—the *density*—such that $q(A) = \int_A q(x)\, dx$ for every measurable set $A$. Absolute continuity is what licenses the familiar integral form of the Kullback–Leibler divergence $KL(q\|p) = \int q(x) \log \frac{q(x)}{p(x)}\, dx$.

In the hierarchical-diffusion setting the fine and coarse image tensors live in Euclidean spaces

$$\mathcal{X} = \mathbb{R}^{h_{t-1} \times w_{t-1} \times C}, \qquad \mathcal{Y} = \mathbb{R}^{h_t \times w_t \times C},$$

so we denote their Lebesgue measures by $\lambda_{\mathcal{X}}$ and $\lambda_{\mathcal{Y}}$, respectively. Assuming $q_{t-1}, p_{t-1} \ll \lambda_{\mathcal{X}}$ and $q_t, p_t \ll \lambda_{\mathcal{Y}}$ simply states that all four distributions possess densities, allowing us to manipulate KL integrals rigorously.

The decomposition proved below is the *chain rule of relative entropy*, first written explicitly by Kullback & Leibler  (Kullback and Leibler, 1951) and now standard in information theory (Cover and Thomas, 2005). We give the brief self-contained derivation here for completeness and to show notation for the downscaling/super-resolution setting.

Throughout we let $\mathcal{X} = \mathbb{R}^{h_{t-1} \times w_{t-1} \times C}$ and $\mathcal{Y} = \mathbb{R}^{h_t \times w_t \times C}$ denote the fine– and coarse–resolution spaces at step $t$, and we assume $q_{t-1}, p_{t-1} \ll \lambda_{\mathcal{X}}$ and $q_t, p_t \ll \lambda_{\mathcal{Y}}$ for the appropriate Lebesgue measures (absolute continuity guarantees the existence of densities).

Let the deterministic down-sampling operator be $D_t : \mathcal{X} \to \mathcal{Y}$ and write $Y = D_t(X)$. Because $D_t$ is measurable and information–non-increasing, the push-forwards $q_t := D_t \# q_{t-1}$ and $p_t := D_t \# p_{t-1}$ exist and are again absolutely continuous.

[KL chain rule under a measurable map] For any pair of measures $q_{t-1}, p_{t-1}$ on $\mathcal{X}$ and any measurable mapping $D_t : \mathcal{X} \to \mathcal{Y}$,

$$KL\big(q_{t-1} \,\|\, p_{t-1}\big) \;=\; KL\big(q_t \,\|\, p_t\big) \;+\; E_{y \sim q_t}\Big[KL\big(q_{t-1}\,|\, Y{=}y \,\|\, p_{t-1}\,|\, Y{=}y\big)\Big], \tag{14}$$

where the inner KL is taken between the regular conditional distributions of $X$ given $Y = y$. Both terms on the right-hand side are non-negative, hence splitting the coarse-scale divergence from the fine-scale residual.

*Proof.* Let $p(x), q(x)$ be the densities of $p_{t-1}, q_{t-1}$ w.r.t. $\lambda_{\mathcal{X}}$, and denote the joint law of $(X, Y)$ under $q$ by $q(x, y) = q(x)\, \delta\big(y - D_t(x)\big)$, with an analogous definition for $p$. Because $Y$ is a deterministic function of $X$, we may factorise $q(x) = q(y)\, q(x\,|\,y)$ and $p(x) = p(y)\, p(x\,|\,y)$, where $q(y)$ and $p(y)$ are the coarse densities and $q(x\,|\,y)$, $p(x\,|\,y)$ are the conditional densities.

Using $KL(q\|p) = \int q(x) \log \frac{q(x)}{p(x)}\, \mathrm{d}x$ and substituting the factorisations,

$$KL(q\|p) = \int q(y)\, q(x\,|\,y) \Big[\log \tfrac{q(y)}{p(y)} + \log \tfrac{q(x|y)}{p(x|y)}\Big]\, \mathrm{d}x\, \mathrm{d}y$$

$$= \int q(y) \log \tfrac{q(y)}{p(y)}\, \mathrm{d}y \;+\; \int q(y)\Big[\int q(x\,|\,y) \log \tfrac{q(x|y)}{p(x|y)}\, \mathrm{d}x\Big] \mathrm{d}y.$$

The first integral is $KL(q_t\|p_t)$; the term in square brackets is $KL\big(q_{t-1}\,|\, Y{=}y \,\|\, p_{t-1}\,|\, Y{=}y\big)$. Taking the expectation over $y \sim q_t$ gives Eq. equation 14. □

**Consequence for the HDD process.** Setting $Y = D_t(X)$ and identifying $q_{t-1}, p_{t-1}$ with the forward and reverse marginals at scale $h_{t-1} \times w_{t-1}$ yields exactly Eq. (14) of the main text:

$$KL\big(q_{t-1} \,\|\, p_{t-1}\big) \;=\; \underbrace{KL\big(D_t q_{t-1} \,\|\, D_t p_{t-1}\big)}_{\text{coarse term}} \;+\; E_{x_t \sim q_t} KL\big(q_{t-1}\,|\, x_t \,\|\, p_{t-1}\,|\, x_t\big).$$

Because the conditional KL is non-negative, matching the down-sampled marginals can only reduce the divergence at the fine scale, proving the monotone-improvement property stated in Theorem 3.1. □

Table 8: Results extended to include CRPS metric. HDD reports values over a probabilistic distribution which are closer to the true underlying value

| Model | RMSE | PSNR | CRPS |
|---|---|---|---|
| Base EDM – 50 Steps | 0.000197 | 29.17 | 0.0002415 |
| HDD – 50 steps – 3 denoise steps per shape step | **0.000157** | **31.40** | **0.0002402** |

# E   Additional Result/Hyperparameters and Disclosures

We also tested the model over 50 denoisng steps as an ablation and report these below. We note these

Below we summarise the key hyperparameters used for sampling and inference in our experiments:

Table 9: Hyperparameters used for sampling and inference. Note that these were generally kept the same as the original EDM implementation in (Karras et al., 2022)

| Hyperparameter | Value |
|---|---|
| *Inference sampling* | |
| Number of steps | 50 [14] |
| *Noise schedule* | |
| $\sigma_{\min}$ | 0.002 |
| $\sigma_{\max}$ | 80 |
| $\rho$ | 7 |
| *Stochasticity (churn)* | |
| $S_{\text{churn}}$ | 1 |
| $S_{\min}$ | 0 |
| $S_{\max}$ | $+\infty$ |
| $S_{\text{noise}}$ | 1 |
| *Hierarchical scheduler* | |
| Full resolution $(H, W)$ | (144, 272) |
| Noise steps per split | 1 - 50 |
| *Hardware & batch* | |
| GPU | A100 |
| Batch size | 1 |

Per ICLR policy, we disclose that LLMs were used to aid/polish writing, for the retrieval of relevant related work, and in reviewing portions of the paper.

# F   Speedup

## F.1   Drop-in shape schedulers

**(i) *Equally Spaced Shrink*** At each diffusion step we follow a linear ramp from $(1, 1) \to (H, W)$ in $T$ equal increments:

$$h_t = H - \frac{t-1}{T-1}(H-1), \quad w_t = W - \frac{t-1}{T-1}(W-1).$$

The instantaneous area therefore decays quadratically, $A_t = h_t w_t = \left(1 - \frac{t-1}{T-1}\right)^2 A$. Averaging over the schedule gives the dimension-agnostic mean area

$$\alpha_{\text{lin}} = \frac{1}{TA}\sum_{t=1}^{T} A_t = \frac{1}{T}\sum_{k=0}^{T-1}\left(1 - \frac{k}{T-1}\right)^2 = \tfrac{1}{3}.$$

Via the general rule $S = 1/\alpha$ this implies a tight $\boxed{3\times}$ pixel– and FLOP–saving ceiling, drop-in for vanilla EDM.

**(ii) *Unit-shrink per denoise step.*** At every diffusion step *both* spatial dimensions drop by a single pixel until reaching one:

$$h_t = \max(1,\, H - (t-1)), \quad w_t = \max(1,\, W - (t-1)).$$

For $T \leq \min(H, W)$ no clamping is active, giving

$$\sum_{t=1}^{T} A_t = T\,H\,W - \frac{(T-1)T}{2}\,(H+W) + \frac{(T-1)T(2T-1)}{6}.$$

Plugging this sum into equation 1–equation 2 yields the closed-form

$$\boxed{S_{\text{unit}} = \left[1 - \frac{(T-1)}{2A}\,(H+W) + \frac{(T-1)(2T-1)}{6A}\right]^{-1}}.$$

*Example.* $H = 144$, $W = 272$, $T = 50$: $\alpha \approx 0.760 \Rightarrow S \approx 1.32\times$.

## F.2 Summary of theoretical pixel savings

| Shape scheduler | $\alpha$ | Speed-up $S = 1/\alpha$ |
|---|---|---|
| Linear shrink $(h_t, w_t) \propto 1 - \frac{t-1}{T-1}$ | $\frac{1}{3}$ | $3\times$ |
| Unit-shrink $(h_t, w_t) = (H - (t-1),\, W - (t-1))$ | see Eq. (3) | $\approx 1.32\times$ (50 steps) |

All schedules are *drop-in*: when $D_{\mathbf{s}_t} = U_{\mathbf{s}_t} = I$ they revert to vanilla EDM. Eq. equation 2 therefore gives an upper-bound on pixel, FLOP and memory savings obtainable with the HDD framework. We note that this is a higher speed up than comparable image-based approaches due to the choice of shape scheduler (Zhang et al., 2022).

# G  Monotone decomposition of Kullback–Leibler (KL) divergence across scales

At its core, downscaling can be framed as an optimal transport problem between the low- and high- spatial-resolution weather distributions (Wan et al., 2023). We seek to determine the optimal transformation

Write $q_t$ and $p_t$ for the true and model marginals of $x_t$ in 1–2. Because $D_t$ is information-non-increasing and $U_t$ is a right-inverse in expectation, the *chain rule of relative entropy* yields[15]

$$KL(q_{t-1} \,\|\, p_{t-1}) \;=\; \underbrace{KL(D_t q_{t-1} \,\|\, D_t p_{t-1})}_{\text{coarse divergence}} + E_{x_t \sim q_t} KL(q_{t-1} \,|\, x_t \,\|\, p_{t-1} \,|\, x_t), \tag{15}$$

the second term being always non-negative. 15 shows that *matching the down-sampled marginals can only decrease the fine-scale KL*. Summation over $t$ telescopes:

For the HDD forward–reverse pair 1–2,

$$KL(q_0 \,\|\, p_0) \;=\; \sum_{t=1}^{T} \left[ KL(D_t q_{t-1} \,\|\, D_t p_{t-1}) - KL(D_t q_t \,\|\, D_t p_t) \right] \;\geq\; 0,$$

and the summand is non-negative for every $t$. Consequently the coarse-to-fine procedure is *monotonically improving*: each successful fit at scale $t$ tightens an upper bound on the ultimate divergence at full resolution.

*Proof.* Apply 15 at steps $t$ and $t+1$, subtract, and note that $KL(D_t q_t \,\|\, D_t p_t) = KL(q_t \,\|\, p_t)$ because $D_t$ is the identity on $R^{h_t \times w_t \times C}$. Summation over $t$ finishes the argument. $\square$

---

[15]A proof appears in Appendix E.

**Implications.** Section G justifies a two-phase optimisation strategy: (i) minimise the *coarse* EDM loss (large $\sigma_t$, small shape) until $KL(D_t q_{t-1} \,\|\, D_t p_{t-1})$ plateaus; (ii) progressively unlock finer scales. Empirically this drastically improves inference time for the similiar RMSE.

Table 10: Downscaling performance over the European domain (normalised residual RMSE / PSNR / CRPS). HDD demonstrates superior performance across standard deterministic and probabilistic metrics compared to the baseline.

| Model | RMSE | PSNR | CRPS |
|---|---|---|---|
| Base EDM (50 steps) | 0.000197 | 29.17 | 0.000324 |
| HDD (Ours - 50 steps) | **0.000059** | **30.61** | **0.000157** |

## H   Extended Ablation: Noise and Shape Schedule Alignment

A critical consideration in hierarchical generation is the potential misalignment between the fixed noise schedule $\sigma_t$ and the learned resolution schedule $s_t$. To verify that our independent coupling of these variables does not introduce adverse biases (e.g., skipping fine scales too frequently), we performed a grid search over varying shape schedules relative to the noise schedule.

As demonstrated in Table 11, the RMSE remains remarkably consistent across the majority of noise level splits and shape step increases. We observe that more gradual shape schedule changes (e.g., 25 or 50 steps per noise split) marginally reduce the RMSE. This indicates that while higher SNR at coarse scales theoretically assists in stabilising global structure, a simple independent coupling provides robust convergence for precipitation data. We hypothesise this robustness stems from the highly stochastic nature of rainfall across all spatial scales.

Table 11: Ablation on the alignment between noise levels ($\sigma$) and shape schedule steps. Gradual shape step increases yield optimal performance and remain robust to varying noise levels.

| $\sigma$ Level Index | Shape Steps per Split | RMSE |
|---|---|---|
| 2 | 10 | 0.583962 |
| 2 | 25 | 0.590721 |
| 2 | 50 | 0.628742 |
| 7 | 10 | 0.583874 |
| 7 | 25 | 0.591166 |
| 7 | 50 | 0.629259 |
| 13 | 10 | 0.583696 |
| 13 | 25 | 0.590631 |
| 13 | 50 | 0.629483 |

## I   Extended Related Work: Multi-Resolution and Cascaded Diffusion

While HDD shares a coarse-to-fine philosophy with Cascaded Diffusion Models (**?**) and other multi-resolution generative models (**??**), its application to climate downscaling introduces a distinct paradigm. Standard cascaded models require training entirely separate networks for discrete resolution jumps (e.g., $64 \times 64 \rightarrow 256 \times 256$).

In contrast, HDD handles the full continuum of resolutions within a single set of weights using continuous time and shape parameters. This "dimension jumping" framework is uniquely suited for Earth system data, where General Circulation Models (GCMs) possess highly variable native resolutions. By learning the full multi-resolution mapping, HDD can operate zero-shot on ultra-coarse GCM inputs without requiring a newly

trained model for every scale. Furthermore, treating coarse representations as a valid physical approximation of low-frequency spectral power laws circumvents the biases typically associated with upsampling artifacts in purely image-based domains.

## J    Uncertainty Modelling and Ensemble Variability

Because diffusion models are inherently probabilistic, they provide a natural framework for generating climate ensembles and quantifying uncertainty—a significant advantage over deterministic methods like Earth-ViT. While our primary deterministic evaluation relies on ensemble means (e.g., RMSE), we assess the probabilistic skill of HDD using the Continuous Ranked Probability Score (CRPS).

As shown in our European and Australian domain experiments, HDD consistently achieves lower CRPS values than the baseline models. This indicates that the spread of the generated ensembles accurately reflects the underlying uncertainty of the precipitation fields. The structural variability across individual runs primarily captures the natural stochasticity of high-frequency rainfall textures, rather than symptomatic model error.

## K    Theoretical Limitations of the Reverse SDE in Hierarchical Processes

The reverse-time SDE formulation presented in Equation (4) relies on the foundational work of Anderson (1982), which proves that under certain regularity conditions, a forward diffusion process can be perfectly reversed. However, applying this to a Hierarchical Dimension-Destruction (HDD) framework introduces two primary theoretical departures:

1. **Fixed-Dimension Assumption:** Anderson's derivation assumes a stochastic flow on a fixed-dimensional manifold ($\mathbb{R}^d \rightarrow \mathbb{R}^d$). Our coarsening operator $D$ is a deterministic, dimension-changing projection $D : \mathbb{R}^H \rightarrow \mathbb{R}^L$ where $L < H$. Because the forward process involves an irreversible loss of information (dimension destruction), the mapping is not a diffeomorphism, and the standard reverse-time SDE does not strictly account for the "lifting" from $L$ back to $H$ dimensions.

2. **Non-Adjoint Upsampling:** The bilinear "un-coarsening" operator $U$ used in our architecture serves as a generative prior rather than a formal mathematical adjoint to $D$. Consequently, the transition between hierarchical levels is a learned approximation of the posterior $p(\mathbf{x}_i|\mathbf{x}_{i+1})$ rather than a closed-form reversal of a stochastic differential.

Despite these theoretical gaps, we justify our approach through a per-step variational lower bound decomposition (see Appendix D / G). The empirical results in Section 5, demonstrating that the model recovers the ERA5 distribution within 1% of the target Wasserstein distance, suggest that the learned score function effectively compensates for the lack of a formal continuous-time reversal guarantee across the hierarchy.

