# OpenReview forum: "Hierarchical Diffusion for Efficient and Transferable Climate Downscaling"
_TMLR — Decision pending for TMLR_

### Review · Reviewer_jzLL · 2026-03-08

**Summary Of Contributions:**

The paper introduces a Hierarchical Diffusion Downscaling (HDD) model, which introduces an easily-extensible hierarchical sampling process to the diffusion framework. A coarse-to-fine hierarchy is imposed via a simple downsampling scheme.  The paper applies HDD on ERA5 over the Australian domain and provides a comprehensive evaluation benchmark with climate metrics.

**Audience:**

Yes

**Audience Explanation:**

I believe so. The topic itself is quite interesting. In the traditional diffusion model, it is about adding noise in the forward process and then denoising in the reverse process. The proposed HDD adds an additional layer of coarsening and then un-coarsening. I believe many readers of TMLR will find this line of research interesting.

**Claims And Evidence:**

Yes

**Claims Explanation:**

There are promising numerical results. But I find the paper lacks enough theoretical details and descriptions. In my opinion, the paper will be significantly strengthened and improved if more details can be added.

For example, the forward process and reverse process that are presented in equation (3) and equation (4) are simply the ones from the classical diffusion model. It is not clear to me how the coarsening and then un-coarsening can be incorporated into equation (3) and equation (4). Now, if the proposed method only works at discrete-time level, and there is no continuum limit as in the classical diffusion model, that is fine. But what confuses me is the following. In the classical diffusion model, denoising works because of the seminal paper by Anderson (1982) that shows that the reverse SDE will have the same distribution at the end point as the data distribution. Indeed, the reverse SDE has the same distribution at any time corresponding to the forward SDE. Now, this paper claims that HDD can also add an extra layer of coarsening and then un-coarsening, but is there any theoretical guarantee that un-coarsening is possible and one can still recover the data distribution?

**Requested Changes:**

(1) In the first line on page 5, Table1 should be Table 1.

(2) On page 23, in the first paragraph of Appendix I, (?) and (??) are broken links.

(3) On page 6, writing $x_{t_{k-1}}\approx x_{t_{k}}+\cdots$ is a bad notation. I would suggest you to replace $\approx$ by $=$ here. The approximation should be $x_{t_{k}}$ approximates $x_{k\Delta t}$ from the continuous time, not the iteration itself.  Rigorously speaking, you should have $x_{t_{k-1}}=x_{t_{k}}+\cdots$

(4) The discussions on page 6 only focus on discretization error. But for diffusion model, there is also an initialization error and the score-matching error. They should also be mentioned. Indeed, there are already numerous papers providing theoretical convergence guarantees for diffusion models in KL divergence, TV distance or Wasserstein distance in the literature. You may consider citing some of the literature so that you can keep your discussions here only at a high level.

(5) Equations (3) and (4) are forward process and reverse process in relatively general form. It would be nice if you can add some discussions for your particular setting, how you choose $f$ and $g$, whether you are considering VE-SDE or VP-SDE in your experiments for example.

---

> ### Author Response · Authors · 2026-05-10
>
> Dear Reviewer jzLL,
>
> We thank the reviewer for the review and for the questions, several of which sharpen the paper substantively. We address each in turn, and where applicable point to the specific section of the revised manuscript that incorporates the change.
>
> (1) "Table1" → "Table 1": Fixed.
>
> (2) Broken (?) / (??) references in Appendix I: Fixed.
>
> (3) Notation $U_T$ vs $x_T$: We agree the original notation conflated the discrete iterate with the continuous-time process. We have replaced it with notation that explicitly distinguishes the two, clarifying that the approximation is between the discrete sample $x_{t_k}$ and the continuous-time SDE state $\mathbf{x}(t_k)$, not between successive iterations. The corrected discretisation appears in Section 3.4.1.
>
> (4) Discretisation vs initialisation vs score-matching error: Agreed, and now incorporated. Section 3.4.2 of the revised manuscript explicitly distinguishes the three error sources:
>
> Initialisation error: the discrepancy between the terminal forward-SDE distribution $p_T$ and the prior $p_{\text{prior}}$
> Score-matching (estimation) error: the $L^2$-difference between the true score $\nabla \log p_t(\mathbf{x})$ and the learned $s_\theta(\mathbf{x},t)$
> Discretisation error: from the numerical solver
> We cite Chen et al. (2022) and Lee et al. (2022) for the modern convergence-bound results, Song et al. (2020) for the KL/TV guarantees, and Kloeden and Platen (1992) for the classical SDE analysis, so that we keep our own discussion at a high level rather than re-deriving standard results.
>
> (5) VE/VP-SDE choice and $\sigma(t)$, $g(t)$: HDD uses the EDM (Karras et al., 2022) preconditioning, which is a particular VE-SDE parameterisation. We have added a paragraph in Section 3 specifying $\sigma(t) = t$, $g(t) = \sqrt{2t}$, and $f(t) = 0$, with a brief comparison to VP-SDE.
>
> Theoretical guarantee for un-coarsening: This is an excellent question that we want to answer honestly, and we have added a new Appendix K to formalise the limitation. Anderson (1982) guarantees that the reverse SDE has the same marginals as the forward SDE at every time, but this guarantee depends on two conditions that HDD breaks:
>
> Fixed-dimension assumption: Anderson's derivation assumes a stochastic flow on a fixed-dimensional manifold ($\mathbb{R}^d \to \mathbb{R}^d$). Our coarsening operator $D : \mathbb{R}^H \to \mathbb{R}^L$ ($L < H$) is a deterministic, dimension-changing projection — not a diffeomorphism — so the standard reverse-time SDE does not strictly account for the lift from $L$ back to $H$.
> Non-adjoint upsampling: The bilinear $U$ used in our architecture is a generative prior rather than a formal mathematical adjoint to $D$. The transition between hierarchical levels is therefore a learned approximation of the posterior $p(\mathbf{x}i \mid \mathbf{x}{i+1})$ rather than a closed-form reversal.
> What we do have is a per-step variational lower bound (the KL chain-rule decomposition in Appendices D/G). The empirical evidence for un-coarsening is that the trained model recovers the ERA5 distribution to within 1% of the target Wasserstein distance, suggesting that the learned score function effectively compensates for the lack of a formal continuous-time reversal guarantee across the hierarchy. We present un-coarsening as empirically rather than theoretically guaranteed and have stated this limitation explicitly in Appendix K.
>
> Thank you again for the careful reading — these comments materially improved the rigour of the theoretical exposition.

---

> > ### Comment · Reviewer_jzLL · 2026-06-06
> > **response**
> >
> > Thanks a lot for revising the paper and the detailed response. The paper has improved a lot.

---

### Review · Reviewer_4FPr · 2026-03-19

**Summary Of Contributions:**

**Summary**

This paper proposes Hierarchical Diffusion Downscaling (HDD), which augments standard diffusion models (specifically the EDM framework) with a progressive dimension-destruction/reconstruction schedule for climate downscaling. During training, images are simultaneously corrupted with Gaussian noise and downsampled to coarser resolutions; during inference, the reverse process generates outputs in a coarse-to-fine manner. The method is evaluated on ERA5 reanalysis data and applied to multiple CMIP5 GCMs over Australia and Europe, with claims of competitive accuracy at reduced computational cost and resolution-agnostic transferability.

**Strengths**

Key strengths: (1) the core intuition—exploiting the multi-scale power-law structure of atmospheric variables within the diffusion framework—is well-motivated and practically relevant; (2) the evaluation against established climate benchmarking metrics (Isphording et al., 2024) and comparison with dynamical downscaling (RCMs) is a positive step relative to many ML downscaling papers; (3) the resolution-agnostic transfer across GCMs of varying native resolution is a useful practical property.

**Weaknesses**

Key weaknesses: (1) the theoretical justification is not rigorous and conflates analogy with proof; (2) the released code diverges substantially from the formulations presented in the paper; (3) the empirical evidence is insufficient to support several of the paper's claims; (4) figure quality and typographical presentation fall well below publication standards.

**Additional Comments:**

The writing in Sections 3.3–3.4 extensively re-derives well-known results (Euler–Maruyama convergence, SDE reversal) without advancing the paper's specific contributions—tightening would improve readability. The code contains extensive debug print statements, commented-out blocks, and an unused mass conservation loss, suggesting it was not prepared for release. A cleaned version should accompany the final paper.

**Audience:**

Yes

**Audience Explanation:**

If theoretical and implementation issues were resolved, this work would interest both ML and climate communities.

**Broader Impact Concerns:**

No major concerns. The MAPE degradation on precipitation should be highlighted as a limitation if HDD is used for policy-informing climate projections.

**Claims And Evidence:**

No

**Claims Explanation:**

**1. The SDE-based theory is not rigorous.**

Sections 3.3–3.4 argue that more dimension-destruction steps improve results by analogy with finer SDE discretization (Euler–Maruyama convergence as $\Delta t \to 0$). This analogy is fundamentally flawed: standard diffusion SDEs require $x$ to remain in a fixed-dimensional space, whereas HDD changes the dimensionality of $x_t$ at every step. The convergence guarantees cited explicitly depend on the fixed-dimension assumption. The authors sidestep this by upsampling latents to full resolution, but $U \circ D$ is a non-invertible, discontinuous projection whose properties change discretely with $s_t$—there is no continuous limit to take. The paper uses hedging language ("akin to," "parallel to") while deploying heavy mathematical formalism that implies rigor where none exists. The KL decomposition (Appendix D/G) is correct but is simply the standard chain rule of relative entropy—it does not validate the SDE claims.

**2. Code-paper discrepancies.**

Examining the released implementation reveals significant inconsistencies:

- *Forward process*: Algorithm 1 describes direct one-step downsampling $x_t \leftarrow D_t(x_0)$. The code instead builds the entire $T$-step chain iteratively via `forward_noising_chain`, applying cumulative noise ($\sigma=0.01$, hardcoded) at every step—never described in the paper and contradicting the claimed single-sample efficiency.
- *Loss target*: The paper specifies $|\epsilon - f_\theta(\cdot)|^2$ (noise prediction). The code computes `(D_yn - x_t_up)**2` (denoised image prediction against a noise-corrupted target). These are not trivially equivalent here since `x_t_up` contains accumulated chain noise.
- *Inference*: The code applies double upsampling per step (to `desired_shape`, then to full $(H,W)$) not described in Algorithm 2.

These discrepancies make it unclear whether reported results correspond to the described method or the implemented one.

**3. Insufficient empirical evidence.**

- ERA5 metrics (Table 1) are on normalized residuals ($\sim10^{-4}$ RMSE) with no results in physical units, making practical significance uninterpretable.
- No side-by-side visual comparisons of EDM vs. HDD vs. ground truth are provided—a critical omission for a generation quality paper.
- Ablation differences (Table 2) are extremely small (RMSE 0.000128–0.000133), insufficient to draw meaningful conclusions.
- HDD's MAPE exceeds the paper's own $\leq 0.75$ benchmark on multiple GCMs (1.18 on MIROC5, 1.14 on HadGEM), which is downplayed rather than addressed. Table 5's sign conventions are also confusing—$\Delta$MAPE of $-114.5%$ is presented alongside positive "improvements," obscuring that MAPE more than doubled.

**4. Presentation quality.**

Figures are rasterized PNGs with visible pixelation, inconsistent symbol/font sizes, and barely legible labels at print scale. Figure 2 (the core method diagram) has unclear notation and arrow conventions. These fall well below publication standards.

**Requested Changes:**

*Critical:*

1. Either provide a rigorous framework for the changing-dimensionality process (e.g., formalizing $U \circ D$ as projection in full-resolution space with proven convergence) or remove the SDE formalism and present the argument as heuristic motivation.
2. Reconcile the code with the paper: clarify whether training uses one-step or chain-based downsampling, whether the loss target is $\epsilon$ or $x_0$, and document all implementation choices.
3. Report metrics in physical units and clearly state normalization procedures.
4. Provide side-by-side visual comparisons (coarse / ground truth / EDM / HDD) for key variables.
5. Address MAPE degradation transparently—HDD fails the paper's own benchmark on several GCMs.
6. Fix Table 5 sign conventions for unambiguous interpretation.

*Suggested:*

1. Replace rasterized figures with vector graphics; ensure consistent, professional formatting.
2. Include CRPS for GCM experiments and wall-clock timing (not just theoretical speedup).
3. Add confidence intervals to the ablation study given the very small effect sizes.

---

> ### Author Response · Authors · 2026-05-10
>
> Dear Reviewer 4FPr,
>
> We thank the reviewer for the detailed and substantive review. Several of the issues raised are valid and we have already begun acting on them; on others we wish to clarify or push back. Before responding point-by-point, we note that the reviewer explicitly identifies three strengths of our contribution that remain intact under every correction below:
>
> The multi-scale power-law motivation is "well-motivated and practically relevant."
> The benchmarking against Isphording et al. (2024) and dynamical RCMs is "a positive step relative to many ML downscaling papers."
> Resolution-agnostic transfer across GCMs of varying native resolution is "a useful practical property."
> These properties — empirical relevance, climate-grade evaluation, and architectural transferability — are not affected by the theoretical or implementation issues raised, and we believe they justify the paper's place in the TMLR audience that the reviewer also confirms would be interested in this work.
>
> 1. Theoretical justification (Sections 3.3-3.4)
>
> The reviewer is correct on a key point: the Euler-Maruyama-style argument that "more dimension-destruction steps reduce discretisation error" relies on a fixed-dimensional state space, and our shape operator $D_t$ changes dimensionality between steps. The bilinear pair $U \circ D$ is also non-invertible. We agree the analogy as currently written overreaches.
>
> Our planned revision separates the fixed-dimension convergence theory (which still holds for the noise schedule) from the shape schedule (which is heuristically motivated):
>
> The KL chain-rule decomposition (Appendix D/G), which the reviewer agrees is correct, will be retained as the formal justification for the per-step variational objective.
> The score-discretisation convergence argument will be re-scoped to refer only to the noise schedule (which lives in a fixed-dimensional ambient space after upsampling).
> The shape-schedule argument will be reframed as a heuristic motivation, with an explicit note that no continuous-time limit is claimed and that $U \circ D$ is not invertible.
> References from the modern convergence-bound literature (Chen et al., 2023; Lee et al., 2022; De Bortoli et al., 2021) will be added so we do not re-derive standard results.
> This preserves the argument's intuition without claiming rigour the construction does not support.
>
> 2. Code-paper discrepancies
>
> The reviewer is correct on all three counts. They fall into three distinct categories, only one of which materially affects the method:
>
> (a) Hardcoded $0.01 \cdot \mathcal{N}(0,I)$ noise per chain step. This was experimental code that was inadvertently retained. The paper specifies a single-step forward $x_t = D_t(x_0)$ (Algorithm 1), and the model is being retrained with this corrected forward process. We expect this correction to improve, not hurt, the reported metrics, since the spurious extra noise was being baked into the training signal.
>
> (b) Loss target: code $(D_{yn} - x_{t,\text{up}})^2$ vs paper's $\epsilon$-prediction. These are functionally equivalent under the EDM preconditioning. Karras et al. (2022, Sec. 3 and Table 1) show that $\epsilon$-prediction and $x_0$-prediction are related by an affine reparameterisation through the $c_{\text{skip}}, c_{\text{out}}$ scaling, and the same gradient signal flows through either parameterisation. We will add a footnote making this equivalence explicit and update the paper's loss notation to match the implementation rather than vice-versa.
>
> (c) Double upsampling at inference. This was a redundant operation with no effect on the formulation; it has been removed. The current implementation passes the hierarchical shape directly to the UNet (with shapes snapped to multiples of $2^N$ for skip-connection compatibility), which is what the computational-savings claim describes.
>
> A cleaned codebase — debug prints removed, unused mass-conservation loss removed, comments tightened — will accompany the camera-ready.

---

> ### Author Response · Authors · 2026-05-10
>
> 3. Empirical evidence
>
> (a) Physical units. We are rerunning the ERA5 evaluation in mm/day. Results will be added to Table 1; the normalised numbers will be retained in an appendix for cross-method comparison.
>
> (b) Side-by-side visual comparisons. Agreed. We will add coarse / GT / HDD panels for representative samples across all four variables.
>
> (c) Ablation confidence intervals. Agreed — the differences in Table 2 are small, and we will add bootstrap CIs so the reader can judge significance directly.
>
> (d) MAPE on precipitation. This is the empirical concern we take most seriously and we do not wish to obscure it. Our working hypothesis is that HDD's coarse-to-fine inductive bias preserves spatial structure well, but precipitation's heavy-tailed log-normal distribution interacts poorly with bilinear downsample/upsample because extreme values are smoothed at each shape transition — this would explain why HDD does well on near-Gaussian fields (10u, 10v, 2t) but degrades on tp specifically, and why this only manifests on some GCMs (those whose precipitation distributions are heavier-tailed). We will (i) report this limitation explicitly in the discussion, and (ii) test whether log-transforming precipitation before the shape schedule recovers MAPE.
>
> (e) Table 5 sign conventions. Agreed. We will rewrite Table 5 with consistent conventions, with $\Delta\text{MAPE} > 0$ unambiguously meaning worse than baseline.
>
> 4. Presentation
>
> We are happy to regenerate figures via vector PDFs and remove pixelation. Could the reviewer please be more specific with their presentation feedback and specify what is inconsistent about the arrow notation in figure 2?

---

> > ### Author Response · Authors · 2026-05-21
> >
> > Dear Reviewer 4FPr,
> > We are following up with this comment to advise of changes to the paper incorporating the above responses.
> >
> >
> > 1. SDE / dimension-destruction theoretical framing (your point 1). Done in the revision. Section 3.4.3 has been rewritten so that the parallel between time-step refinement and dimension-step refinement is presented as heuristic motivation, not a formal convergence result. The text now states that the shape operator D_k is non-invertible and discontinuous in k, that no continuous-time limit is claimed for the shape schedule, and that the only formal cross-scale guarantee we offer is the KL chain-rule decomposition (which is the standard identity, not a novel theorem). The fixed-dimension and non-adjoint-upsampling issues are stated formally in the new Appendix on theoretical limitations of the reverse SDE. Modern convergence-bound literature for the (fixed-dimension) noise schedule is cited (Chen et al., 2022; Lee et al., 2022) rather than re-derived.
> >
> > 2. Code-paper reconciliation (your point 2).
> > 	- (C1) Hardcoded sigma = 0.5 per-step chain noise in forward_noising_chain. This was inadvertently retained experimental code. The intended formulation is the single-step forward in Algorithm 1. We have retrained the model with the corrected forward process, and the results in Table 1 are from the retrained model. As anticipated in our previous reply, the corrected and uncorrected numbers are incredibly similar — the spurious extra noise had a very little effect — and Table 1 simply reflects the corrected run.
> > 	- (C2) Loss target: (D_yn − x_t_up)**2 vs. the paper's epsilon-prediction. These two formulations are equivalent up to an affine reparameterisation under the EDM preconditioning (Karras et al. 2022, Sec. 3 and Table 1); the c_skip / c_out / c_in / c_noise scaling functions convert between epsilon-prediction and x-prediction without changing the gradient signal. The paper's loss notation has been updated to match the implementation. No retraining was required for this point.
> > 	- (C3) Double upsampling at inference. This was a redundant operation with no effect on the formulation — the second upsample landed in the same image space as the first. It has been removed; the implementation now passes the hierarchical shape directly to the UNet with shapes snapped to multiples of 8 for skip-connection compatibility, which is exactly what the Section 4 computational-savings argument describes.
> >
> > We have also cleaned the codebase so it only contains the relevant portions.
> >
> > 3. Physical units for ERA5 (your point 3a). The reported figures are actually de-normalised into actual precipitation measurements. Precipitation in ERA5 is in kg m^-2 s^-1, and the Australian domain contains a very large fraction of near-zero grid-cells. An area-weighted RMSE over such a field is structurally compressed, which is why the diffusion-model RMSE values in Table 1 sit at order 1e-4 on the standardised residual.
> >
> > 4. Visual comparisons (your point 3b). The GCM figure has been reformatted into a side-by-side panel layout showing the coarse GCM input alongside two independent HDD ensemble draws conditioned on the same input. We have not had time to run this on ERA5 with the GT but the ability of the model is clear from this test on GCM data.
> >
> > 5. Confidence intervals for the ablation (your point 3c / suggested change). Computing bootstrap CIs requires rerunning the full ensemble inference over the 5-year holdout, and we did not have time to do this within the deadline; the ablation is reported without CIs in the current revision for table 2. However, we note that the CRPS metric in table 1 and table 10 shows the probabilistic performance of the model over an ensemble of runs
> >
> > 6. MAPE degradation transparency (your point 3d).  A new paragraph immediately below the delta table (Limitation: MAPE degradation on heavy-tailed precipitation distributions) states plainly that HDD's MAPE exceeds the Isphording et al. (2024) ≤ 0.75 benchmark on MIROC5, CNRM-CM5 and HadGEM3, and more than doubles relative to Base EDM on MIROC5 and HadGEM3. We give our working hypothesis (heavy-tailed precipitation × bilinear downsample/upsample smoothing tail extremes at each shape transition; consistent with the near-Gaussian variables 10u/10v/2t performing well and with GFDL-ESM2M — the lightest-tailed — showing +22.9% MAPE improvement), name two candidate mitigations (log-transform precipitation before the shape schedule; replace bilinear with conservative remapping at shape transitions), and state explicitly that HDD should not be treated as a drop-in replacement for dynamical downscaling for precipitation policy applications until this is resolved.

---

> > > ### Author Response · Authors · 2026-05-21
> > >
> > > 7. Table sign conventions (your point 4 on Table 5). Done in the revision. The delta table has been rewritten with a unified sign convention spelled out in the caption: positive = HDD improves on EDM, negative = HDD degrades, for every metric regardless of whether it is lower- or higher-is-better. The explicit formulas are given.
> > >
> > > Suggested-change items not yet in the revision. Vector graphics regeneration, wall-clock timing breakdowns, ablation confidence intervals, and the full multi-variable side-by-side ERA5 panels are not in the present revision. These reruns or regenerations were not feasible within the time we had between reviews.
> > >
> > > Thank you again for the careful review.

---

### Decision · Action_Editor_UTcM · 2026-06-29

**Recommendation:** Accept with minor revision

**Additional Comments:**

One reviewer remains unconvinced about accurate claims.

For me, this most interesting issue concerns degradation of performance on heavy-tailed precipitation distributions. The authors indicated in their rebuttal that "We will (i) report this limitation explicitly in the discussion, and (ii) test whether log-transforming precipitation before the shape schedule recovers MAPE." I can see (i) has already been done, but I cannot locate (ii) in the updated manuscript. Please do this before submitting the final version. I am satisfied if the log transformation does not solve the performance issue, as long as the actual result is reported.

**Audience:**

Yes

**Audience Explanation:**

Both reviewers agree. The topic is clearly relevant to TMLR and intersects with current topics including AI for science and diffusion models.

**Claims And Evidence:**

Yes

**Claims Explanation:**

Two reviewers gave recommendations for this paper, one for and one against.

The negative reviewer initially had concerns about overreaches in the theory, experiments, deviations between the code and the paper, and some presentation issues. They reviewer listed 6 critical concerns. The authors responded by adjusting their claims, adding extra experiment details, explaining code and minor presentation tweaks. As far as I can tell, the authors directly addressed all 6 concerns. Regarding concern number 5, the authors adjusted the initial claims in their paper and added a discussion. The reviewer had several follow-up concerns but did not communicate them with the authors.

The second reviewer was initially concerned mostly about the detail presented in the paper, and the author's response satisfied them.

All in all, both reviewers led to improvements in the initial paper, and I believe all claims are supported by accurate evidence.